# How Data Influence Changes in Training? Time-Varying Influence Measurement

## Abstract

Existing data influence analyses are static, measuring the global, cumulative influence of training data on fully trained models while leaving dynamic changes during training a black box. We propose Time-varying Influence Measurement (TIM), the first framework measuring how data influence changes during training. TIM operates on arbitrary local windows, estimating how removing a training point within a window affects model parameters, and then projects these parameter deviations onto task-relevant functional responses (e.g., test loss) via query vectors. We establish theoretical error bounds under non-convex and non-converged conditions. Experiments show that: 1) TIM estimates loss changes more accurately than prior methods and closely matches Leave-One-Out (LOO) retraining; 2) Data influence is time-varying, exhibiting different patterns including Early Influencers, Late Bloomers, Stable Influencers, and Highly Fluctuating patterns; 3) Global or longer windows are not necessarily better, as small-window TIM achieves better performance in corrupted data identification while reducing cost by 95%.

## 1 Introduction

Modern machine learning systems are trained on massive datasets of different quality. Understanding *which* training data matter, *when* they matter during training, and *how* they affect the model, is important for building trustworthy, efficient, and interpretable Artificial Intelligence (AI) systems. However, most existing influence analyses Koh & Liang (2017); Ghorbani & Zou (2019) are *static*: they estimate a single, aggregated/average influence of training data on a fully trained model, leaving how data influence changes during training unexplored.

Current methods have fundamental limitations for measuring time-varying influence dynamics. Leave-One-Out retraining (LOO) provides a gold standard but is computationally infeasible at scale. Influence Functions (IF) Koh & Liang (2017); Guo et al. (2021) assume an optimal point, which is fragile in non-convex, non-converged scenarios Basu et al. (2021); Bae et al. (2022). Custom scoring methods compute task-related scores during training but fail to quantify actual loss changes. For example, Shapley Value methods Ghorbani & Zou (2019) ensure fairness in data valuation tasks by averaging marginal contributions, but only provide *expected utility* rather than true loss changes in a specific run. TracIn Pruthi et al. (2020) similarly uses gradient inner products as a proxy, rather than quantifying true loss changes. These methods fundamentally cannot capture how data influence changes during training, which we term time-varying influence.

It is challenging to measure time-varying influence. First, it is computationally intensive, requiring comparison of model states with and without each data point across training while the model continuously evolves rather than remaining fixed at convergence. Second, a new theoretical framework is needed for analyzing intermediate model states during training, as existing methods rely on model convergence Basu et al. (2021). Third, it is difficult to connect training data, parameter updates, and functional responses (e.g., test loss, predictions) during training, as this requires tracking high-dimensional, time-dependent parameter-to-function mappings.

To address these challenges, we propose Time-varying Influence Measurement (TIM), a novel framework that efficiently quantifies how training data influence changes during training. TIM operates within arbitrary windows of the training process rather than only analyzing the final model. Specifically, TIM first estimates how excluding training data within a window affects model parameters, then projects these parameter deviations onto task-relevant query vectors to measure functional

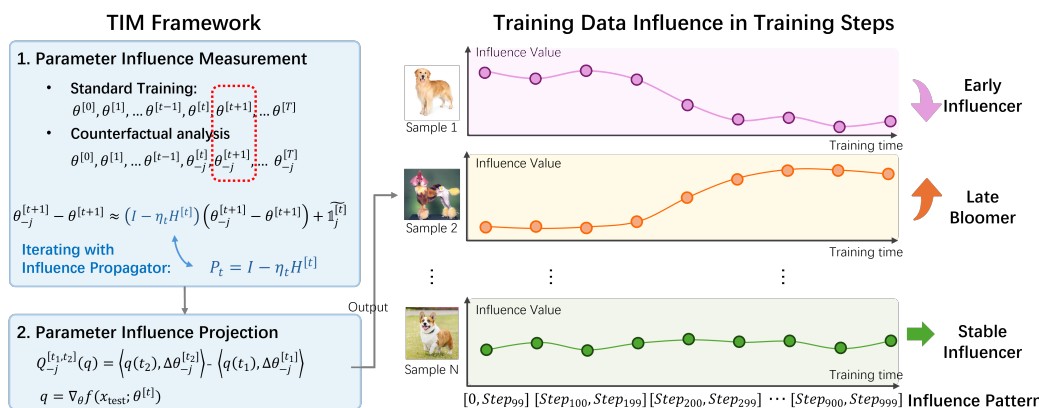

Figure 1: Overview of the Time-Varying Influence Measurement (TIM) Framework. TIM's two-stage approach: *(1) Parameter Influence Measurement* uses recursive estimation with Influence Propagator $P_t = I - \eta_t H^{[t]}$ to track parameter deviations; *2) Parameter Influence Projection* maps parameter changes to functional responses (e.g., test loss) via query vectors $q$. By analyzing data influence across different training windows, TIM enables fine-grained temporal influence analysis.

responses (e.g., test loss). This projection mechanism provides an interpretable and computationally efficient connection between parameter changes and functional responses. Figure 1 illustrates the TIM framework.

Our experiments reveal three key insights: 1) The influence of training data is time-varying. Different data have different patterns: Early Influencers, Late Bloomers, Stable Influencers, and Highly Fluctuating (Figure 4). 2) Global scope or longer analysis windows do not mean better accuracy for data influence analysis. Small-window TIM achieves superior performance with 95% cost reduction (Table 6). 3) TIM matches LOO accuracy while significantly outperforming existing baselines.

Overall, the contributions of this paper are summarized as follows.

- We propose TIM, the first framework to measure time-varying data influence over training windows. TIM connects parameter changes to functional responses via query vectors, enabling understanding how different data contribute to learning at different training stages.
- We establish theoretical error bounds robust to non-convergence and non-convexity without restrictive assumptions required by existing methods (Appendix B).
- Extensive evaluations demonstrate that TIM outperforms baselines while matching LOO accuracy, reveals distinct time-varying influence patterns, and shows that small-window analysis achieves superior performance with cost reduction.

## 2 RELATED WORKS

Data influence analysis methods can be broadly categorized into 1) LOO approximation methods, which estimate true LOO retraining influence, and 2) *custom scoring* methods that provide heuristic utility (e.g., outlier detection, data pruning) without approximating retraining loss. TIM belongs to the first category, offering LOO estimates with an upper error bound (Appendix B).

LOO retraining is the gold standard for measuring data influence, but is prohibitively expensive, motivating the development of efficient approximation methods. Influence Functions (IFs) Koh & Liang (2017) and recent extensions Guo et al. (2021); Schioppa et al. (2022); Choe et al. (2024); Grosse et al. (2023) approximate LOO influence on the final converged model using Taylor approximations, but their accuracy degrades in non-convex settings or under incomplete convergence Schioppa et al. (2023); Basu et al. (2021). More importantly, recent analyses Bae et al. (2022) highlight that IFs fail to approximate true LOO due to warm-start bias and proximal mismatch. SGD-influence Hara et al. (2019) quantifies loss changes with a recursive approximation of parameter differences of the whole

training process, while DVEmb Wang et al. (2025b) uses a similar recursive framework to study the influence of the data position in the training sequence, but their approach lacks theoretical analysis and shows poor experimental results. Existing retraining approximation methods focus on explaining the final trained model, without addressing how influence changes across training windows.

Custom influence score methods offer computational efficiency by focusing on practical proxies for influence. These methods are highly effective for tasks like data valuation and data pruning/cleansing, as these tasks do not require precise loss changes. Shapley value approaches Ghorbani & Zou (2019); Jia et al. (2021); Wang et al. (2024; 2025a) and domain-specific adaptations Schoch et al. (2022); Sun et al. (2023); Wang & Jia (2023); Li & Yu (2023) prioritize theoretical fairness by averaging over run-specific stochasticity (e.g., data order). They only provide the expected contribution of a data point to a learning algorithm, not its actual influence in a training process. OFA Li & Yu (2024) accelerates convergence with optimized sampling, while Data-OOB Kwon & Zou (2023) avoids retraining by reusing out-of-bag, but it is restricted to bagging ensembles. TracIn Pruthi et al. (2020) is a representative method that measures influence by accumulating gradient products across checkpoints. For data pruning/cleansing, GraNd and EL2N scores Paul et al. (2021) prune data by ranking data according to the expected norm of their loss gradients. YOCO He et al. (2023) extends EL2N with balanced dataset construction. MoSo Tan et al. (2024) prunes data using the inner product between the data's gradient and the average gradient. CGSV Xu et al. (2021) and cosine similarity methods Fung et al. (2018); Xia et al. (2024) analyze gradient alignment at individual iterations. These methods do not estimate LOO retraining loss, but are validated by downstream tasks.

TIM advances LOO approximation by providing the first framework to estimate LOO retraining within any training window, capturing the time-varying influence of training data.

## 3 PRELIMINARIES AND PROBLEM FORMULATION

**Preliminaries.** Let $\mathcal{Z} = \mathcal{X} \times \mathcal{Y}$ denote the space of observations, where $\mathcal{X} \subseteq \mathbb{R}^d$ is the input space and $\mathcal{Y}$ is the output space. Given a training dataset $D = \{z_i\}_{i=1}^N$ of i.i.d. observations $z_i = (x_i, y_i) \in \mathcal{Z}$, a model $f : \mathcal{X} \times \Theta \to \mathcal{Y}$ parameterized by $\theta \in \Theta \subseteq \mathbb{R}^p$, and a loss function $\ell : \mathcal{Z} \times \Theta \to \mathbb{R}$, we formulate the learning problem as $\hat{\theta} = \arg\min_{\theta \in \Theta} \frac{1}{N} \sum_{i=1}^N \ell(z_i; \theta)$.

Stochastic Gradient Descent (SGD) is a representative method for solving this optimization problem. Most data influence analysis methods Koh & Liang (2017); Pruthi et al. (2020); Hara et al. (2019) are built upon SGD, and we also adopt SGD for fair comparison. Let $g(z; \theta) = \nabla_\theta \ell(z; \theta)$, and the initialization parameters is $\theta^{[0]}$. At each step $t$, a mini-batch $S_t \subseteq \{1, ..., N\}$ is sampled and SGD iteratively updates the parameters according to:

$$\theta^{[t+1]} = \theta^{[t]} - \frac{\eta_t}{|S_t|} \sum_{i \in S_t} g(z_i; \theta^{[t]}), \quad 0 \le t \le T-1, \tag{1}$$

where $\eta_t$ is the learning rate at step $t$ and $T$ is the total number of SGD steps.

**Problem Formulation.** Fix a window $[t_1, t_2]$ with $0 \le t_1 < t_2 \le T$. Given a training process $\{\theta^{[t]}\}_{t=0}^T$, let $\{\theta_{-j}^{[t]}\}_{t=0}^T$ be the LOO trajectory obtained by running the same SGD with shared initialization $\theta_{-j}^{[0]} = \theta^{[0]}$ but excluding $z_j$ from updates. We aim to quantify the time-varying influence of $z_j$ in $[t_1, t_2]$ on: 1) parameter-trajectory deviation $\Delta\theta_{-j}^{[t_1,t_2]} = (\theta_{-j}^{[t_2]} - \theta_{-j}^{[t_1]}) - (\theta^{[t_2]} - \theta^{[t_1]})$; 2) functional responses, such as test loss $\Delta\ell_{-j}^{[t_1,t_2]} = (\ell_{\text{test}}(\theta_{-j}^{[t_2]}) - \ell_{\text{test}}(\theta_{-j}^{[t_1]})) - (\ell_{\text{test}}(\theta^{[t_2]}) - \ell_{\text{test}}(\theta^{[t_1]}))$.

## 4 TIME-VARYING INFLUENCE MEASUREMENT (TIM) FRAMEWORK

### 4.1 PARAMETER INFLUENCE MEASUREMENT

This section defines parameter influence as trajectory deviation, an approximation to the LOO retraining influence, measuring the difference in the learning path with and without $z_j$ over $[t_1, t_2]$.

To formalize this, we first define a LOO training process where $z_j$ is excluded. This process starts from the same initialization $\theta_{-j}^{[0]} = \theta^{[0]}$, and updates at each step $t$ as:

$$\theta_{-j}^{[t+1]} = \theta_{-j}^{[t]} - \frac{\eta_t}{|S_t|} \sum_{i \in S_t \setminus \{j\}} g(z_i; \theta_{-j}^{[t]}), \quad 0 \leq t \leq T - 1. \tag{2}$$

This allows us to formally define the parameter influence of $z_j$ over $[t_1, t_2]$ as the difference between these two trajectories:

$$\Delta\theta_{-j}^{[t_1, t_2]} = (\theta_{-j}^{[t_2]} - \theta_{-j}^{[t_1]}) - (\theta^{[t_2]} - \theta^{[t_1]}), \tag{3}$$

where $(\theta_{-j}^{[t_2]} - \theta_{-j}^{[t_1]})$ denotes the parameter change on the LOO trajectory when $z_j$ is excluded during $[t_1, t_2]$, and $(\theta^{[t_2]} - \theta^{[t_1]})$ denotes the change on the original trajectory.

**Recursive Estimation.** Computing $\Delta\theta_{-j}^{[t_1, t_2]}$ directly requires costly model retraining. Instead, we develop a recursive estimation approach that tracks parameter deviations step-by-step. The standard SGD update for step $t$ is:

$$\theta^{[t+1]} = \theta^{[t]} - \frac{\eta_t}{|S_t|} \sum_{i \in S_t} g(z_i; \theta^{[t]}). \tag{4}$$

When excluding data $z_j$, the parameter update becomes:

$$\theta_{-j}^{[t+1]} = \theta_{-j}^{[t]} - \frac{\eta_t}{|S_t|} \sum_{i \in S_t \setminus \{j\}} g(z_i; \theta_{-j}^{[t]}). \tag{5}$$

For step $t$, the difference between the standard update and the update excluding $z_j$ is:

$$\theta_{-j}^{[t+1]} - \theta^{[t+1]} = (\theta_{-j}^{[t]} - \theta^{[t]}) - \frac{\eta_t}{|S_t|} \Big( \sum_{i \in S_t \setminus \{j\}} g(z_i; \theta_{-j}^{[t]}) - \sum_{i \in S_t} g(z_i; \theta^{[t]}) \Big). \tag{6}$$

To handle the gradient differences, we employ a Taylor expansion around $\theta^{[t]}$:

$$g(z_i; \theta_{-j}^{[t]}) - g(z_i; \theta^{[t]}) \approx H_i^{[t]} (\theta_{-j}^{[t]} - \theta^{[t]}), \tag{7}$$

where $H_i^{[t]} = \nabla_\theta^2 \ell(z_i; \theta^{[t]})$ is the Hessian of the loss for $z_i$. In Section 5.1, our experiments show that our method achieves superior accuracy than baselines, even with this approximation.

Averaging Eq. (7) over $S_t$ and defining $H^{[t]} = \frac{1}{|S_t|} \sum_{i \in S_t} \nabla_\theta^2 \ell(z_i; \theta^{[t]})$, we obtain:

$$\frac{1}{|S_t|} \sum_{i \in S_t} (g(z_i; \theta_{-j}^{[t]}) - g(z_i; \theta^{[t]})) \approx H^{[t]} (\theta_{-j}^{[t]} - \theta^{[t]}). \tag{8}$$

**Influence Propagation.** Substituting Eq. (8) into Eq. (6) and approximating $H^{[t]} \approx H_{-j}^{[t]}$ (see the full derivation in Appendix A.1), we derive the core recurrence relation:

$$\theta_{-j}^{[t+1]} - \theta^{[t+1]} \approx (I - \eta_t H^{[t]})(\theta_{-j}^{[t]} - \theta^{[t]}) + \mathbf{1}_{j \in S_t} \frac{\eta_t}{|S_t|} g(z_j; \theta^{[t]}), \tag{9}$$

where $\mathbf{1}_{j \in S_t}$ is an indicator function that equals 1 if $j \in S_t$, otherwise 0.

We define $P_t := I - \eta_t H^{[t]}$ as **Influence Propagator**, which characterizes how influence propagates through training steps. This recurrence reveals that parameter deviation at step $t + 1$ comprises two components: 1) historical influence, which is the previous deviation $(\theta_{-j}^{[t]} - \theta^{[t]})$ propagated forward and modulated by $P_t$; 2) instantaneous influence, which is new contribution $\tilde{\mathbf{1}}_j^{[t]} = \mathbf{1}_{j \in S_t} \frac{\eta_t}{|S_t|} g(z_j; \theta^{[t]})$ from $z_j$ at the current step.

**Final Estimator.** Recursively applying the influence propagation Eq. (9) over the training window $[t_1, t_2]$ and accounting for accumulated influence before $t_1$ (complete derivation in Appendix A.2), we obtain our estimator:

$$\widehat{\Delta\theta}_{-j}^{[t_1, t_2]} = \left( \prod_{k=t_1}^{t_2-1} P_k - I \right) \left( \sum_{t=0}^{t_1-1} \left( \prod_{k=t+1}^{t_1-1} P_k \right) \tilde{\mathbf{1}}_j^{[t]} \right) + \sum_{t=t_1}^{t_2-1} \left( \prod_{k=t+1}^{t_2-1} P_k \right) \tilde{\mathbf{1}}_j^{[t]}. \tag{10}$$

To validate the robustness of this estimator, we provide a theoretical error bound in Appendix B. Our analysis confirms the error holds for non-convex settings without requiring model convergence, and is controlled by key training parameters like the learning rate and Hessian smoothness, making it broadly applicable to modern deep learning. Our experiments in Section 5 also confirm this result and show superior accuracy compared to baselines.

## 4.2 Influence Projection using Query Vectors

While Section 4.1 quantifies how training data affects *model parameters*, it does not directly reveal the influence on *model functional responses*, such as test loss, predictions, or feature importance. To bridge this gap, we introduce a projection-based mechanism that connects parameter changes to functional responses through query vectors. This approach is grounded in a well-established principle that small parameter changes lead to approximately linear changes in model outputs Hampel (1974); Hara et al. (2019). It is the foundation of influence function Koh & Liang (2017), and has been empirically validated in various deep learning scenarios Park et al. (2023); Ilyas et al. (2022). This enables us to predict functional changes from parameter deviations via directional derivatives, which serve as our query vectors.

A query vector $q(t) \in \mathbb{R}^p$ encodes the sensitivity of a specific model response to parameter changes. It defines a direction in parameter space, and the inner product $\langle q(t), \Delta\theta \rangle$ measures how much the parameter change $\Delta\theta$ projects onto this response-relevant direction.

**Definition 4.1** (Query-based TIM). Let $q : [0, T] \to \mathbb{R}^p$ be a query function that maps time $t$ to a query vector $q(t) \in \mathbb{R}^p$. The query-based TIM for a training data $z_j$ over the time window $[t_1, t_2]$ is defined as:

$$Q_{-j}^{[t_1, t_2]}(q) = \langle q(t_2), \Delta\theta_{-j}^{[t_2]} \rangle - \langle q(t_1), \Delta\theta_{-j}^{[t_1]} \rangle, \tag{11}$$

where $\langle \cdot, \cdot \rangle$ denotes the standard inner product in $\mathbb{R}^p$, and $\Delta\theta_{-j}^{[t]} = \Delta\theta_{-j}^{[0,t]}$ for brevity.

This definition provides a versatile framework for analyzing various model functional responses (e.g., test loss, predictions) through different $q$. For example, using the test loss gradient, $q(t) = \nabla_\theta \ell(z_{\text{test}}; \theta^{[t]})$, we have:

$$\begin{aligned}
Q_{-j}^{[t_1, t_2]}(q) &= \langle \nabla_\theta \ell(z_{\text{test}}; \theta^{[t_2]}), \Delta\theta_{-j}^{[t_2]} \rangle - \langle \nabla_\theta \ell(z_{\text{test}}; \theta^{[t_1]}), \Delta\theta_{-j}^{[t_1]} \rangle \\
&\approx [\ell(z_{\text{test}}; \theta_{-j}^{[t_2]}) - \ell(z_{\text{test}}; \theta_{-j}^{[t_1]})] - [\ell(z_{\text{test}}; \theta^{[t_2]}) - \ell(z_{\text{test}}; \theta^{[t_1]})].
\end{aligned} \tag{12}$$

This directly approximates the change in test loss difference caused by excluding $z_j$ during $[t_1, t_2]$. Additionally, we can use $q = \nabla_\theta f(x_{\text{test}}; \theta^{[t]})$ to measure prediction changes, $q(t) = \nabla_x \nabla_\theta \ell(z_{\text{test}}; \theta^{[t]})$ for feature importance, and $q = e_i$ (standard basis vector) for individual parameter importance. Appendix C details how TIM can be applied to diverse functional responses. In this work, we focus on test loss as a representative case, since it directly reflects model generalization and serves as a key benchmark in prior influence analyses.

## 4.3 Implementation of TIM

TIM efficiently computes data influence $Q_{-j}^{[t_1, t_2]}(q)$ by running a single backward sweep over the targeted window and using Hessian–vector products (HVPs) only. Table 1 compares TIM with baselines across computational complexity and robustness metrics. TIM achieves superior efficiency while maintaining robustness to non-convex, non-converged training dynamics, making it practical for large-scale applications. Detailed algorithms and implementations are provided in Appendix D.

Table 1: Comprehensive comparison of different data influence analysis methods

| Aspect | LOO | IF | TracIn | LAVA | DVEmb | TIM |
|---|---|---|---|---|---|---|
| Computation Cost | $O(NC_{train})$ | $O(p^3)$ | $O(KNp)$ | $O(NMd)$ | $O(|S_t|T\tilde{p}^2)$ | $O(w|S_t|p)$ |
| Storage Cost | $O(p)$ | $O(p^2)$ | $O(KNp)$ | $O(NM)$ | $O(|S_t|T\tilde{p})$ | $O(w(|S_t|+p))$ |
| Robustness to Non-convergence | Yes | No | Yes | Yes | Yes | Yes |
| Robustness to Non-convexity | Yes | No | Yes | Yes | Yes | Yes |
| Approximation LOO | Yes | Yes | No | No | Yes | Yes |

$T$ = total steps, $p$ = param dimension, $\tilde{p}$ = projection dim., $d$ = projection dim., $|S_t|$ = batch size, $K$ = # checkpoints, $w$ = window size.

## 5 EXPERIMENTS

We evaluate TIM by first evaluating its accuracy (Sections 5.1 and 5.2), analyzing its ability to capture evolving data influence (Section 5.3, secpattern and Appendix F.4), and demonstrating the significant benefits of TIM's unique time-varying perspective in downstream applications (Section 5.5). Full specifications and baseline method descriptions are provided in Appendix F.1.

### 5.1 ACCURACY OF INFLUENCE MEASUREMENT

We evaluate TIM's accuracy by comparing its influence estimates against the LOO gold standard across two scenarios: 1) **global analysis** over the entire training trajectory $[0, T]$, and 2) **local analysis** over temporal windows $[t_1, t_2]$. We compare TIM against Influence Functions (IF) Koh & Liang (2017), LAVA Just et al. (2023), and DVEmb Wang et al. (2025b) using four complementary metrics: Pearson and Spearman correlations (linear and monotonic consistency), Kendall's $\tau$ (ordinal ranking), and Jaccard similarity on the top 30% most influential points.

**Comparison of Global Analysis.** We first examine each method's ability to approximate LOO loss changes over the whole training trajectory $[0, T]$ for MNIST-DNN across 20 epochs. TIM consistently achieves near-perfect agreement with LOO retraining (correlations $> 0.9$), significantly higher than other baselines. In contrast, IF and DVEmb achieve only moderate agreement, while LAVA fails with near-zero correlation due to its custom scoring rather than retraining-based influence estimates. These results validate TIM's recursive estimation approach and demonstrate that TIM can accurately estimate global influence.

Table 2: Correlation with LOO for **global influence analysis**.

| Method | Pearson | Spearman | Kendall's Tau | Jaccard (Top 30%) |
|---|---|---|---|---|
| IF Koh & Liang (2017) | 0.75±0.14 | 0.70±0.17 | 0.52±0.14 | 0.52±0.19 |
| DVEmb Wang et al. (2025b) | 0.58±0.12 | 0.49±0.29 | 0.35±0.21 | 0.34±0.20 |
| LAVA Just et al. (2023) | -0.07±0.10 | 0.03±0.10 | 0.02±0.07 | 0.22±0.06 |
| TIM | **0.96**±0.03 | **0.94**±0.06 | **0.83**±0.08 | **0.78**±0.15 |

**Comparison of Local Window Analysis.** We next examine how well methods capture time-varying influence within local windows. Since IF, LAVA, and DVEmb only produce global influence, we construct their local estimates by differencing the loss between $[0, t_2]$ and $[0, t_1]$. While this is not their original design, it provides the fairest possible adaptation for local settings; otherwise, these methods cannot be applied. In contrast, TIM directly estimates influence within a window $[t_1, t_2]$. We evaluate on 21 consecutive windows $[e, e+1]$ ($e = 0, \ldots, 20$) and report the average correlation with LOO. TIM again shows superior performance with both high accuracy, whereas IF and DVEmb remain moderate, and LAVA remains ineffective. This confirms that TIM achieves accurate influence estimates within local windows.

Table 3: Correlation with LOO for **local analysis** (averaged over 21 per-epoch windows $[e, e+1]$).

| Method | Pearson | Spearman | Kendall's Tau | Jaccard (Top 30%) |
|---|---|---|---|---|
| IF Koh & Liang (2017) | 0.70±0.02 | 0.65±0.02 | 0.48±0.01 | 0.50±0.03 |
| DVEmb Wang et al. (2025b) | 0.56±0.02 | 0.48±0.05 | 0.35±0.03 | 0.35±0.04 |
| LAVA Just et al. (2023) | -0.06±0.01 | 0.05±0.01 | 0.04±0.02 | 0.21±0.01 |
| TIM | **0.95**±0.01 | **0.93**±0.01 | **0.81**±0.02 | **0.77**±0.02 |

### 5.2 SCALABILITY TO LARGE-SCALE MODELS

**Corrupted data detection.** To evaluate TIM on large-scale models, we conduct experiments on BERT-IMDB sentiment classification with 50% randomly flipped labels. We measure the precision of identifying corrupted data among the worst X% ranked points (X = 20, 30, 40, 50) across training epochs (Figure 2).

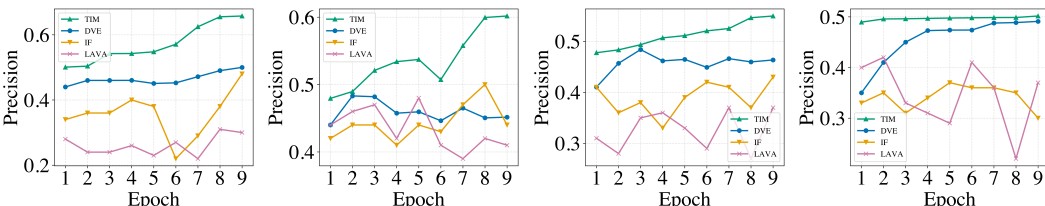

Figure 2: Precision of corrupted data detection on BERT-IMDB (left to right: 20%, 30%, 40%, 50% selection thresholds).

Across all thresholds, TIM consistently achieves the highest precision and shows steady improvements over training. The advantage is most evident under strict settings ($X = 20$), where detection is most difficult, but TIM also maintains strong performance as the threshold expands to 50%. DVEmb and IF deliver moderate performance, while LAVA remains consistently lowest, which is consistent with the findings in the benchmark study OpenDataVal Jiang et al. (2023) on noisy-label detection. These results confirm TIM's robustness and scalability, demonstrating that it remains effective under extreme noise and is well-suited for large-scale, non-convex models such as BERT.

**Convergence Acceleration through Data Pruning.** Beyond corrupted data detection, we evaluate its effectiveness in accelerating model convergence through data pruning. Using the same BERT-IMDB setup with 50% corrupted labels, we identify the 10% worst-performing data points using different methods and remove them from training. We then measure training loss convergence when training on the pruned datasets (Figure 3).

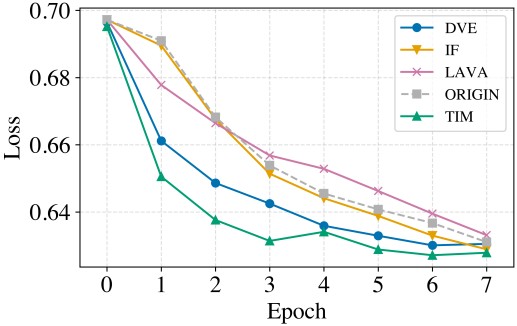

Figure 3: Convergence comparison after pruning corrupted data identified by different methods.

TIM achieves the most significant acceleration in convergence, consistently reaching lower training loss than both the original corrupted dataset and all baseline methods. DVEmb and IF provide moderate improvements, while LAVA yields negligible gains and occasionally slows convergence due to unstable pruning. These results highlight TIM's practical value in identifying truly harmful training data, enabling more efficient optimization in noisy, large-scale training settings.

## 5.3 PATTERNS OF DATA INFLUENCE DYNAMICS

While existing methods provide static data influence analysis, our study reveals that training data have different time-varying influence patterns during training. To uncover this, we compute data influence on test loss at each epoch throughout training. This results in a time series of influence values for each data point, capturing its evolving influence on test loss. Full implementation details are provided in Appendix F.2.

As model training progresses, test loss naturally decreases. This causes raw influence values to shrink over time for all training data, masking how relative influence evolves during training. To address this, we standardize the computed influence values at each epoch, preserving relative importance while

removing the global declining scale effect. We then fit linear trends to each standardized time series to analyze the long-term trend. By analyzing trend direction, statistical significance, and temporal variability, we identify four distinct influence patterns (Early Influencers, Late Bloomers, Stable Influencers, and Highly Fluctuating) as shown in Figure 4.

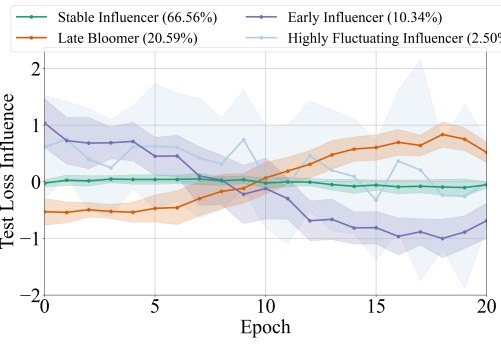

- **Early Influencers**: High early influence that diminishes over time.
- **Late Bloomers**: Influence increases as training progresses.
- **Stable Influencers**: Consistent influence throughout training.
- **Highly Fluctuating Influencers**: Influence varies significantly throughout training.

Figure 4: Time-varying influence patterns on MNIST training using DNNs

We further analyzed the pattern distribution across datasets and models, as shown in Table 4. These results show several key insights. 1) Time-varying influence patterns exist across all dataset-model combinations. This nature underscores the limitations of static influence analysis. 2) The presence of Early Influencers and Late Bloomers reveals that models selectively emphasize different training data at different stages. 3) Pattern distributions vary significantly across model architectures and data modalities, emphasizing the necessity of dynamic influence analysis approaches.

Table 4: Distribution of influence dynamic patterns across datasets and models (percentage)

| Model | Dataset | Stable Influencer | Early Influencers | Late Bloomers | Highly Fluctuating |
|---|---|---|---|---|---|
| LR | Adult | 64.75±7.20 | 11.67±3.27 | 20.15±5.87 | 3.42±1.82 |
| | 20News | 85.94±5.38 | 1.17±1.28 | 5.57±1.26 | 7.32±4.24 |
| | MNIST | 80.16±12.10 | 0.79±0.96 | 10.78±9.35 | 8.27±3.36 |
| | EMNIST | 75.49±8.40 | 0.70±0.53 | 13.77±6.77 | 10.04±2.75 |
| DNN | Adult | 97.91±2.66 | 0.313±1.12 | 1.00±1.55 | 0.78±0.89 |
| | 20News | 79.03±7.78 | 8.44±4.11 | 11.41±3.90 | 1.13±0.83 |
| | MNIST | 66.56±13.26 | 10.34±4.65 | 20.59±9.44 | 2.50±0.93 |
| | EMNIST | 78.16±14.48 | 7.09±7.678 | 7.47±9.87 | 7.28±3.55 |
| CNN | MNIST | 83.76±19.91 | 0.34±0.42 | 11.74±16.60 | 4.15±3.94 |
| | EMNIST | 86.50±7.50 | 1.87±5.15 | 1.59±3.91 | 10.03±2.48 |

## 5.4 Pattern-Specific Accuracy

We conducted a pattern-specific performance analysis comparing TIM with LOO as ground truth using the MNIST with DNNs. We divided training data into Stable, Early, Late, and Highly Fluctuating according to Section 5.3, and report correlations between TIM and LOO within each pattern cluster. Table 5 presents the comparative results across multiple evaluation metrics.

Table 5: Pattern-specific accuracy of TIM approximating LOO.

| Data Pattern | Pearson | Spearman | Kendall's Tau | Jaccard (Top 30%) |
|---|---|---|---|---|
| **Stable Influencers** | 0.95±0.03 | 0.96±0.03 | 0.87±0.05 | 0.82±0.12 |
| **Early Influencers** | 0.94±0.04 | 0.98±0.01 | 0.92±0.03 | 0.89±0.07 |
| **Late Bloomers** | 0.98±0.02 | 0.98±0.02 | 0.90±0.05 | 0.85±0.10 |
| **Highly Fluctuating** | 0.76±0.18 | 0.72±0.18 | 0.63±0.21 | 0.52±0.34 |

The pattern-specific analysis reveals three key findings. First, TIM achieves excellent approximation accuracy for LOO across patterns. All correlations for Stable, Early, and Late patterns exceed 0.94, with Late Bloomers showing the highest correlation (0.98). Second, TIM remains positively correlated with LOO even for Highly Fluctuating patterns. Third, Stable/Early/Late patterns exhibit low variance, while Highly Fluctuating patterns show high variance, suggesting sensitivity to seeds and requiring smoothing or multi-seed aggregation. Influence dynamics across training stages are detailed in Appendix F.4.

## 5.5 DIFFERENT WINDOW SELECTION

To investigate how different windows affect data influence measurement, we evaluated TIM's ability to identify corrupted data on MNIST binary classification task (digits '1' and '7'). We randomly selected and flipped labels for 5%, 10%, 15%, and 20% of training data (corresponding to 12, 25, 38, and 51 data points). For each corruption level, we trained models over 20 epochs and computed influence using different temporal windows: **full**-training TIM, and **epoch**-window TIM (first, middle, and last epochs). We compare against LOO retraining as the gold standard. Table 6 shows each method's precision, defined as correctly identified flipped labels among the top-$k$ most negatively influential data points, where $k$ is the actual number of corrupted samples.

Table 6: Identification of corrupted data

| Flipped | Model | LOO | Full-training TIM | First-epoch TIM | Mid-epoch TIM | **Last-epoch TIM** |
|---|---|---|---|---|---|---|
| | LR | **10.94 ± 0.90** | **10.94 ± 0.90** | 10.56 ± 1.22 | 10.88 ± 0.78 | 10.88 ± 0.78 |
| 12 | DNN | 8.81 ± 1.98 | 9.06 ± 1.85 | 8.25 ± 2.33 | 8.88 ± 2.09 | **9.38 ± 1.98** |
| | CNN | 10.44 ± 1.32 | 10.50 ± 1.32 | 8.75 ± 2.11 | 10.69 ± 1.16 | **11.06 ± 1.32** |
| | LR | **23.50 ± 1.00** | **23.50 ± 1.00** | 22.56 ± 1.54 | **23.50 ± 1.06** | 23.38 ± 1.00 |
| 25 | DNN | 19.94 ± 3.77 | 20.75 ± 3.01 | 20.31 ± 2.78 | 20.50 ± 3.22 | **21.31 ± 3.77** |
| | CNN | 21.75 ± 3.11 | 21.81 ± 3.11 | 18.44 ± 4.37 | 22.19 ± 2.81 | **23.56 ± 3.11** |
| | LR | **36.06 ± 1.14** | **36.06 ± 1.14** | 35.38 ± 1.62 | 35.69 ± 1.69 | 35.13 ± 1.14 |
| 38 | DNN | 32.50 ± 3.72 | 32.81 ± 3.47 | 32.19 ± 3.40 | 32.56 ± 3.61 | **33.31 ± 3.72** |
| | CNN | 34.19 ± 4.17 | 34.19 ± 4.17 | 29.75 ± 5.93 | 34.56 ± 3.98 | **36.31 ± 4.17** |
| | LR | **48.69 ± 1.16** | **48.69 ± 1.16** | 47.94 ± 1.52 | 46.56 ± 3.12 | 42.94 ± 1.16 |
| 51 | DNN | 43.94 ± 5.20 | 45.31 ± 3.29 | 44.13 ± 3.64 | 45.19 ± 3.30 | **45.56 ± 5.20** |
| | CNN | 46.25 ± 4.35 | 46.19 ± 4.33 | 41.50 ± 7.66 | 47.13 ± 3.35 | **48.69 ± 4.35** |

First, TIM closely matches the LOO gold standard across all corruption levels, providing reliable detection without retraining. Second, for convex models (LR), gradient dynamics remain stable, making full-training TIM only marginally better. Third, for non-convex models (CNN, DNN), last-epoch TIM achieves the best or near-best detection while reducing computation by 95% compared to full-training TIM, since its window length is one epoch versus the entire training of 20 epochs. This demonstrates that smaller temporal windows (Last-epoch TIM) can be more efficient and sometimes more effective than analyzing the entire training trajectory (Full-training TIM), challenging the assumption that longer analysis windows necessarily yield better influence estimates.

## 6 CONCLUSION

We presented TIM, a framework for measuring how training data influence evolves over time. Unlike static methods, TIM approximates LOO within arbitrary training windows and projects parameter deviations onto functional responses via query vectors. Our analysis establishes error bounds that hold under non-convex and non-converged conditions, ensuring theoretical robustness. Experiments show that TIM matches LOO accuracy, reveals distinct temporal patterns, and enables practical gains such as corrupted data detection and accelerated convergence, while reducing computation by 95%.

ETHICS STATEMENT

This work complies with the ICLR Code of Ethics. All datasets are publicly available and widely used benchmarks. No human subjects, private data, or sensitive attributes are involved. We anticipate no direct ethical risks beyond those generally associated with machine learning research.

REPRODUCIBILITY STATEMENT

Our implementation and scripts are available at `https://anonymous.4open.science/r/TIM-DE8E/`. Section 5 and Appendix F.1 describe datasets, model architectures, and hyperparameters. Proofs of theoretical results appear in Appendix B, and metric definitions are detailed in Appendix F.1. Together, these ensure full reproducibility.

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

# A  DERIVATION OF THE PARAMETER INFLUENCE ESTIMATOR

## A.1  DERIVATION OF THE ONE-STEP RECURRENCE RELATION (EQ. 9)

We start from Eq. (6), which establishes the relationship:

$$\theta_{-j}^{[t+1]} - \theta^{[t+1]} = (\theta_{-j}^{[t]} - \theta^{[t]}) - \frac{\eta_t}{|S_t|}\Big(\sum_{i\in S_t\setminus\{j\}} g(z_i;\theta_{-j}^{[t]}) - \sum_{i\in S_t} g(z_i;\theta^{[t]})\Big) \tag{13}$$

$$= (\theta_{-j}^{[t]} - \theta^{[t]}) - \frac{\eta_t}{|S_t|}\Big(\sum_{i\in S_t\setminus\{j\}} g(z_i;\theta_{-j}^{[t]}) - \sum_{i\in S_t\setminus\{j\}} g(z_i;\theta^{[t]}) - \mathbf{1}_{j\in S_t}g(z_j;\theta^{[t]})\Big) \tag{14}$$

$$= (\theta_{-j}^{[t]} - \theta^{[t]}) - \frac{\eta_t}{|S_t|}\Big(\sum_{i\in S_t\setminus\{j\}} g(z_i;\theta_{-j}^{[t]}) - \sum_{i\in S_t\setminus\{j\}} g(z_i;\theta^{[t]})\Big) + \frac{\eta_t}{|S_t|}\mathbf{1}_{j\in S_t}g(z_j;\theta^{[t]}) \tag{15}$$

$$= (\theta_{-j}^{[t]} - \theta^{[t]}) - \frac{\eta_t}{|S_t|}\Big(\sum_{i\in S_t\setminus\{j\}} (g(z_i;\theta_{-j}^{[t]}) - g(z_i;\theta^{[t]}))\Big) + \frac{\eta_t}{|S_t|}\mathbf{1}_{j\in S_t}g(z_j;\theta^{[t]}), \tag{16}$$

where $\mathbf{1}_{j\in S_t}$ is an indicator function that equals 1 if $j \in S_t$, otherwise 0.

Using Eq. (7), we have:

$$\sum_{i\in S_t\setminus\{j\}} (g(z_i;\theta_{-j}^{[t]}) - g(z_i;\theta^{[t]})) \approx \sum_{i\in S_t\setminus\{j\}} H_i^{[t]}(\theta_{-j}^{[t]} - \theta^{[t]}), \tag{17}$$

Using Eq. (8) and Assumption (A4) detailed in Appendix B, we have:

$$\sum_{i\in S_t\setminus\{j\}} H_i^{[t]}(\theta_{-j}^{[t]} - \theta^{[t]}) \approx |S_t|H_{-j}^{[t]}(\theta_{-j}^{[t]} - \theta^{[t]}) \approx |S_t|H^{[t]}(\theta_{-j}^{[t]} - \theta^{[t]}). \tag{18}$$

Combining Eq. (17) and Eq. (18), we have:

$$\sum_{i\in S_t\setminus\{j\}} (g(z_i;\theta_{-j}^{[t]}) - g(z_i;\theta^{[t]})) \approx |S_t|H^{[t]}(\theta_{-j}^{[t]} - \theta^{[t]}). \tag{19}$$

Applying Eq. (19) to Eq. (16), we have the final result:

$$\theta_{-j}^{[t+1]} - \theta^{[t+1]} = (\theta_{-j}^{[t]} - \theta^{[t]}) - \frac{\eta_t}{|S_t|}\sum_{i\in S_t\setminus\{j\}} (g(z_i;\theta_{-j}^{[t]}) - g(z_i;\theta^{[t]})) + \frac{\eta_t}{|S_t|}\mathbf{1}_{j\in S_t}g(z_j;\theta^{[t]}) \tag{20}$$

$$\approx (\theta_{-j}^{[t]} - \theta^{[t]}) - \frac{\eta_t}{|S_t|}(|S_t|H^{[t]}(\theta_{-j}^{[t]} - \theta^{[t]})) + \frac{\eta_t}{|S_t|}\mathbf{1}_{j\in S_t}g(z_j;\theta^{[t]}) \tag{21}$$

$$= (\theta_{-j}^{[t]} - \theta^{[t]}) - \eta_t H^{[t]}(\theta_{-j}^{[t]} - \theta^{[t]}) + \frac{\eta_t}{|S_t|}\mathbf{1}_{j\in S_t}g(z_j;\theta^{[t]}) \tag{22}$$

$$= (I - \eta_t H^{[t]})(\theta_{-j}^{[t]} - \theta^{[t]}) + \frac{\eta_t}{|S_t|}\mathbf{1}_{j\in S_t}g(z_j;\theta^{[t]}). \tag{23}$$

This derivation confirms the correctness of Eq. (9), including the last term.

## A.2  FROM THE RECURRENCE RELATION TO THE FINAL INFLUENCE ESTIMATOR (EQ. 28)

We start from

$$\theta_{-j}^{[t+1]} - \theta^{[t+1]} \approx (I - \eta_t H^{[t]})(\theta_{-j}^{[t]} - \theta^{[t]}) + \mathbf{1}_{j\in S_t}\frac{\eta_t}{|S_t|}g(z_j;\theta^{[t]}), \tag{24}$$

where $\mathbf{1}_{j\in S_t}$ is an indicator function that equals 1 if $j \in S_t$, otherwise 0. Recursively applying Eq. (9) over the training window $[t_1, t_2]$:

$$\theta_{-j}^{[t_2]} - \theta^{[t_2]} \approx P_{t_2-1}P_{t_2-2}\ldots P_{t_1}(\theta_{-j}^{[t_1]} - \theta^{[t_1]}) + \sum_{t=t_1}^{t_2-1} P_{t_2-1}P_{t_2-2}\ldots P_{t+1}\tilde{\mathbf{1}}_j^{[t]}, \tag{25}$$

where $\tilde{\mathbf{1}}_j^{[t]} = \mathbf{1}_{j \in S_t} \frac{\eta_t}{|S_t|} g(z_j; \theta^{[t]})$. Combining Eq. (3) and Eq. (25), we can get:

$$\Delta\theta_{-j}^{[t_1,t_2]} \approx \left(\prod_{k=t_1}^{t_2-1} P_k - I\right)(\theta_{-j}^{[t_1]} - \theta^{[t_1]}) + \sum_{t=t_1}^{t_2-1}\left(\prod_{k=t+1}^{t_2-1} P_k\right)\tilde{\mathbf{1}}_j^{[t]}. \tag{26}$$

We use Eq. (26) for the interval $[0, t_1]$ with $\theta_{-j}^{[0]} = \theta^{[0]}$ to get:

$$\theta_{-j}^{[t_1]} - \theta^{[t_1]} \approx \sum_{t=0}^{t_1-1}\left(\prod_{k=t+1}^{t_1-1} P_k\right)\tilde{\mathbf{1}}_j^{[t]}. \tag{27}$$

Substituting Eq. (27) into Eq. (26), we obtain our final approximation:

$$\Delta\theta_{-j}^{[t_1,t_2]} \approx \left(\prod_{k=t_1}^{t_2-1} P_k - I\right)\left(\sum_{t=0}^{t_1-1}\left(\prod_{k=t+1}^{t_1-1} P_k\right)\tilde{\mathbf{1}}_j^{[t]}\right) + \sum_{t=t_1}^{t_2-1}\left(\prod_{k=t+1}^{t_2-1} P_k\right)\tilde{\mathbf{1}}_j^{[t]}, \tag{28}$$

# B    ESTIMATION ERROR ANALYSIS WITHOUT CONVEXITY ASSUMPTIONS

**Theorem B.1** (Error Bound for TIM Parameter Change). *Let $\Delta\theta_{-j}^{[t_1,t_2]}$ be the true influence of excluding data $z_j$ on the model parameters over the interval $[t_1, t_2]$ during SGD training. Let $\widehat{\Delta\theta}_{-j}^{[t_1,t_2]}$ be its approximation using TIM. Under the following assumptions:*

    *(A1) Lipschitz Continuity of Gradient: The gradient $\nabla\ell(z_i; \theta)$ is Lipschitz continuous with constant $L_g$: $\|\nabla\ell(z_i; \theta_1) - \nabla\ell(z_i; \theta_2)\| \le L_g\|\theta_1 - \theta_2\|$, $\forall \theta_1, \theta_2 \in \Theta, \forall i$.*

    *(A2) Lipschitz Continuity of Hessian: The Hessian $\nabla^2\ell(z_i; \theta)$ is Lipschitz continuous with constant $L_H$: $\|\nabla^2\ell(z_i; \theta_1) - \nabla^2\ell(z_i; \theta_2)\| \le L_H\|\theta_1 - \theta_2\|$, $\forall \theta_1, \theta_2 \in \Theta, \forall i$.*

    *(A3) Learning Rate Bound: The learning rate satisfies $\eta_t \le \frac{1}{L_H}$ for all $t$.*

    *(A4) Hessian Approximation Error: The Hessian approximation error is bounded: $\|H^{[t]} - H_{-j}^{[t]}\| \le \epsilon_H$, $\forall t$, where $H_{-j}^{[t]} = \frac{1}{|S_t\setminus\{j\}|}\sum_{i\in S_t\setminus\{j\}} \nabla^2\ell(z_i; \theta^{[t]})$ is the empirical Hessian over the mini-batch.*

    *(A5) Gradient Norm Bound: For all $\theta \in \Theta$ and all $z_i$: $\|\nabla\ell(z_i; \theta)\| \le G$.*

    *(A6) Parameter Difference Bound: There exists a constant $M > 0$ such that: $\|\theta_{-j}^{[t]} - \theta^{[t]}\| \le M$, $\forall t \in [t_1, t_2]$.*

    *(A7) Bounded Hessian Norm: For all $\theta \in \Theta$ and all $z_i$: $\|\nabla^2\ell(z_i; \theta)\| \le M_H$.*

*Then, the expected estimation error is bounded as follows:*

$$\mathbb{E}\left[\left\|\Delta\theta_{-j}^{[t_1,t_2]} - \widehat{\Delta\theta}_{-j}^{[t_1,t_2]}\right\|\right] \le \frac{\tilde{B}}{M_H}\left(e^{M_H\eta_{\max}(t_2+1)} + e^{M_H\eta_{\max}(t_1+1)} - 2\right) \tag{29}$$

*where: $\eta_{\max} = \max_{t\in[t_1,t_2]}\eta_t$, $\tilde{B} = \frac{L_H M^2}{2} + \epsilon_H M$, $n$ is the total number of data in the dataset.*

*Proof.* **Step 1: Derivation of the Error Update Equation**

Define the error at iteration $t$:

$$e^{[t]} = (\theta_{-j}^{[t]} - \theta^{[t]}) - \widehat{\Delta\theta}_{-j}^{[0,t]} \tag{30}$$

where $\widehat{\Delta\theta}_{-j}^{[0,t]}$ is the approximation of the true parameter change $\Delta\theta_{-j}^{[0,t]}$ using the TIM method.

We aim to derive a recursive relation for $e^{[t]}$ and then bound its expected norm.

Consider the updates for $\theta^{[t]}$, $\theta_{-j}^{[t]}$, and $\widehat{\theta}_{-j}^{[t]}$:

Original SGD Update:

$$\theta^{[t+1]} = \theta^{[t]} - \eta_t\tilde{g}^{[t]}, \quad \tilde{g}^{[t]} = \frac{1}{|S_t|}\sum_{i\in S_t}\nabla\ell(z_i; \theta^{[t]}). \tag{31}$$

Leave-One-Out SGD Update:

$$\theta_{-j}^{[t+1]} = \theta_{-j}^{[t]} - \eta_t\tilde{g}_{-j}^{[t]}, \quad \tilde{g}_{-j}^{[t]} = \frac{1}{|S_t|}\sum_{i\in S_t\setminus\{j\}}\nabla\ell(z_i; \theta_{-j}^{[t]}). \tag{32}$$

Approximate Leave-One-Out Update (TIM Method):

$$\widehat{\theta}_{-j}^{[t+1]} = \widehat{\theta}_{-j}^{[t]} - \eta_t\left(\tilde{g}^{[t]} + H^{[t]}(\widehat{\theta}_{-j}^{[t]} - \theta^{[t]}) - \mathbf{1}_{\{j\in S_t\}}\frac{1}{|S_t|}\nabla\ell(z_j; \theta^{[t]})\right). \tag{33}$$

We derive the error update equation as follows:

$$e^{[t]} - e^{[t-1]} = \eta_{t-1}\delta^{[t-1]}, \tag{34}$$

where:

$$\delta^{[t-1]} = \left( \tilde{g}_{-j}^{[t-1]} - \tilde{g}^{[t-1]} \right) - H^{[t-1]} \widehat{\Delta\theta}_{-j}^{[t-1]} + \mathbf{1}_{\{j \in S_{t-1}\}} \frac{1}{|S_{t-1}|} \nabla\ell(z_j; \theta^{[t-1]}). \tag{35}$$

or equivalently:

$$\delta^{[t]} = \left( \tilde{g}_{-j}^{[t]} - \tilde{g}^{[t]} \right) - H^{[t]} \widehat{\Delta\theta}_{-j}^{[0,t]} + \mathbf{1}_{\{j \in S_t\}} \frac{1}{|S_t|} \nabla\ell(z_j; \theta^{[t]}). \tag{36}$$

**Step 2: Bounding $\|\delta^{[t]}\|$**

We decompose $\delta^{[t]}$ and bound each term:

**1. Difference in Stochastic Gradients:**

$$\tilde{g}_{-j}^{[t]} - \tilde{g}^{[t]} = \frac{1}{|S_t|} \left( \sum_{i \in S_t \setminus \{j\}} \left( \nabla\ell(z_i; \theta_{-j}^{[t]}) - \nabla\ell(z_i; \theta^{[t]}) \right) - \mathbf{1}_{\{j \in S_t\}} \nabla\ell(z_j; \theta^{[t]}) \right). \tag{37}$$

Applying a first-order Taylor expansion to $\nabla\ell(z_i; \theta_{-j}^{[t]})$ for $i \neq j$:

$$\nabla\ell(z_i; \theta_{-j}^{[t]}) - \nabla\ell(z_i; \theta^{[t]}) = \nabla^2\ell(z_i; \theta^{[t]})(\theta_{-j}^{[t]} - \theta^{[t]}) + r_{i,j}^{[t]}, \tag{38}$$

where, by Assumption (A2):

$$\|r_{i,j}^{[t]}\| \leq \frac{L_H}{2} \|\theta_{-j}^{[t]} - \theta^{[t]}\|^2 \tag{39}$$

Thus, we have:

$$\tilde{g}_{-j}^{[t]} - \tilde{g}^{[t]} = \frac{1}{|S_t|} \sum_{i \in S_t \setminus \{j\}} \nabla^2\ell(z_i; \theta^{[t]})(\theta_{-j}^{[t]} - \theta^{[t]}) + r_{i,j}^{[t]} - \mathbf{1}_{\{j \in S_t\}} \nabla\ell(z_j; \theta^{[t]})$$

$$= \frac{1}{|S_t|} \left( \sum_{i \in S_t \setminus \{j\}} r_{i,j}^{[t]} - \mathbf{1}_{\{j \in S_t\}} \nabla\ell(z_j; \theta^{[t]}) \right) + H_{-j}^{[t]}(\theta_{-j}^{[t]} - \theta^{[t]}) \tag{40}$$

**2. Hessian Approximation Error:**

$$\|(H_{-j}^{[t]} - H^{[t]})(\theta_{-j}^{[t]} - \theta^{[t]})\| \leq \epsilon_H \|\theta_{-j}^{[t]} - \theta^{[t]}\| \tag{41}$$

according to Assumption (A4).

**3. Combining Terms:** Substitute the approximations back into $\delta^{[t]}$:

$$\delta^{[t]} = \left( \tilde{g}_{-j}^{[t]} - \tilde{g}^{[t]} \right) - H^{[t]} \widehat{\Delta\theta}_{-j}^{[0,t]} + \mathbf{1}_{\{j \in S_t\}} \frac{1}{|S_t|} \nabla\ell(z_j; \theta^{[t]})$$

$$= \left( \tilde{g}_{-j}^{[t]} - \tilde{g}^{[t]} \right) - H_{-j}^{[t]}(\theta_{-j}^{[t]} - \theta^{[t]}) + \left( H_{-j}^{[t]} - H^{[t]} \right)(\theta_{-j}^{[t]} - \theta^{[t]}) + \mathbf{1}_{\{j \in S_t\}} \frac{1}{|S_t|} \nabla\ell(z_j; \theta^{[t]})$$

$$= \frac{1}{|S_t|} \sum_{i \in S_t \setminus \{j\}} r_{i,j}^{[t]} + (H_{-j}^{[t]} - H^{[t]})(\theta_{-j}^{[t]} - \theta^{[t]}) + H^{[t]}((\theta_{-j}^{[t]} - \theta^{[t]}) - \Delta\widehat{\theta}_{-j}^{[t]})$$

$$= \frac{1}{|S_t|} \sum_{i \in S_t \setminus \{j\}} r_{i,j}^{[t]} + (H_{-j}^{[t]} - H^{[t]})(\theta_{-j}^{[t]} - \theta^{[t]}) + H^{[t]} e^{[t]}. \tag{42}$$

**4. Bounding $\|\delta^{[t]}\|$:**

• **First Term:**

$$\left\| \frac{1}{|S_t|} \sum_{i \in S_t \setminus \{j\}} r_{i,j}^{[t]} \right\| < \frac{L_H M^2}{2}. \tag{43}$$

- **Second Term:**

$$\left\| (H_{-j}^{[t]} - H^{[t]})(\theta_{-j}^{[t]} - \theta^{[t]}) \right\| \leq \epsilon_H M. \tag{44}$$

- **Third Term:**

$$\left\| H^{[t]} e^{[t]} \right\| \leq M_H \|e^{[t]}\|. \tag{45}$$

Combining bounds, we can have:

$$\|\delta^{[t]}\| < \frac{L_H M^2}{2} + \epsilon_H M + M_H \|e^{[t]}\|. \tag{46}$$

**Step 3: Error Update Equation**

Using the error update:

$$e^{[t]} = e^{[t-1]} - \eta_t \delta^{[t-1]}, \tag{47}$$

we have:

$$\|e^{[t]}\| \leq \|e^{[t-1]}\| + \eta_t \|\delta^{[t-1]}\| < \|e^{[t-1]}\| + \eta_t \left( \frac{L_H M^2}{2} + \epsilon_H M + M_H \|e^{[t-1]}\| \right). \tag{48}$$

Define:

$$a_t = 1 + \eta_t M_H, \quad b_t = \eta_t \left( \frac{L_H M^2}{2} + \epsilon_H M \right). \tag{49}$$

Then:

$$\|e^{[t]}\| < a_t \|e^{[t-1]}\| + b_t. \tag{50}$$

**Step 4: Taking Expectations**

Taking expectations over the mini-batch sampling:

$$\mathbb{E}\left[ \|e^{[t]}\| \right] < a_t \mathbb{E}\left[ \|e^{[t-1]}\| \right] + b_t. \tag{51}$$

Define:

$$\tilde{B} = \frac{L_H M^2}{2} + \epsilon_H M. \tag{52}$$

Then:

$$\mathbb{E}\left[ \|e^{[t]}\| \right] < a_t \mathbb{E}\left[ \|e^{[t-1]}\| \right] + \eta_t \tilde{B}. \tag{53}$$

**Step 5: Solving the Recurrence Relation**

Unfolding the recurrence:

$$\mathbb{E}\left[ \|e^{[t]}\| \right] \leq \prod_{k=0}^{t} a_k \cdot \mathbb{E}\left[ \|e^{[0]}\| \right] + \sum_{s=0}^{t} \left( \prod_{k=s+1}^{t} a_k \right) b_s. \tag{54}$$

Since $e^{[0]} = 0$, we have:

$$\mathbb{E}\left[ \|e^{[t]}\| \right] \leq \sum_{s=0}^{t} \left( \prod_{k=s+1}^{t} a_k \right) b_s. \tag{55}$$

Assuming $a_k \leq e^{M_H \eta_{\max}}$, we get:

$$\prod_{k=s+1}^{t} a_k \leq e^{M_H \eta_{\max}(t-s)}. \tag{56}$$

Therefore:

$$\mathbb{E}\left[ \|e^{[t]}\| \right] \leq \tilde{B} \eta_{\max} \sum_{s=0}^{t} e^{M_H \eta_{\max}(t-s)}. \tag{57}$$

Approximating the sum:

$$\mathbb{E}\left[\|e^{[t]}\|\right] \leq \tilde{B}\eta_{\max} \cdot \frac{e^{M_H \eta_{\max}(t+1)} - 1}{e^{M_H \eta_{\max}} - 1}. \tag{58}$$

For small $M_H \eta_{\max}$, $e^{M_H \eta_{\max}} - 1 \approx M_H \eta_{\max}$, yielding:

$$\mathbb{E}\left[\|e^{[t]}\|\right] \leq \frac{\tilde{B}}{M_H}\left(e^{M_H \eta_{\max}(t+1)} - 1\right). \tag{59}$$

Substitute $t$ with $t_1$ and $t_2$ respectively:

$$\mathbb{E}\left[\|e^{[t_2]}\|\right] \leq \frac{\tilde{B}}{M_H}\left(e^{M_H \eta_{\max}(t_2+1)} - 1\right), \tag{60}$$

$$\mathbb{E}\left[\|e^{[t_1]}\|\right] \leq \frac{\tilde{B}}{M_H}\left(e^{M_H \eta_{\max}(t_1+1)} - 1\right). \tag{61}$$

**Step 6: Final Bound**

The estimation error is:

$$\mathbb{E}\left[\left\|\Delta\theta_{-j}^{[t_1,t_2]} - \widehat{\Delta\theta}_{-j}^{[t_1,t_2]}\right\|\right] \leq \mathbb{E}\left[\|e^{[t_2]}\|\right] + \mathbb{E}\left[\|e^{[t_1]}\|\right]$$

$$\leq \frac{\tilde{B}}{M_H}\left(e^{M_H \eta_{\max}(t_2+1)} + e^{M_H \eta_{\max}(t_1+1)} - 2\right) \tag{62}$$

This completes the proof. $\square$

*Remark* B.2. Note that TIM applies to non-converged and non-convex models. The exponential form arises from the recursive nature of error propagation, where each SGD step compounds previous errors multiplicatively. Our analysis is the first to guarantee error bounds for non-converged, non-convex models during any time windows. The bounds are mathematical guarantees for the worst case, and experimental results show that TIM can achieve near-zero errors empirically.

*Remark* B.3. The error bound provides several key insights:

- The error grows at most exponentially with both $t_1$ and $t_2$, highlighting the challenge of long-range influence estimation. The impact of $t_2$ is generally more significant as it represents the end of the time window.

- The Hessian approximation error $\epsilon_H$ directly impacts the overall error, emphasizing the importance of accurate Hessian estimation.

- The maximum learning rate $\eta_{\max}$ affects the error bound exponentially, suggesting that smaller learning rates might help control the estimation error.

- The bound depends on the Lipschitz constants of the gradient and Hessian ($L_g$ and $L_H$), indicating that smoother loss landscapes lead to more reliable influence estimates.

This theorem provides a theoretical foundation for understanding the limitations of influence estimation without assuming convexity and guides practical considerations in its application to large-scale machine learning problems.

# C  TIM TOOLKIT

This appendix provides a practical analysis and implementation guide for common query vectors used in TIM. These include gradient-based, prediction-based, and parameter-specific directions that enable targeted investigation into model functional responses.

## C.1  TIM FOR LOSS VALUE

**Theorem C.1** (TIM for Loss Value). *Given a loss function $\ell(z; \theta)$, a time window $[t_1, t_2]$, a training data $z_j$, and a test data $z_{test}$, the TIM with query function $q(t) = (\nabla_\theta \ell(z_{test}; \theta^{[t]})$ can be approximated as:*

$$Q_{-j}^{[t_1, t_2]}(q) \approx [\ell(z_{test}; \theta_{-j}^{[t_2]}) - \ell(z_{test}; \theta_{-j}^{[t_1]})] - [\ell(z_{test}; \theta^{[t_2]}) - \ell(z_{test}; \theta^{[t_1]})], \tag{63}$$

*where $\theta_{-j}^{[t]}$ denotes the model parameters at time $t$ when trained without data $z_j$, and $\theta^{[t]}$ denotes the parameters when trained with all data.*

*Proof.* We begin with the definition of the query-based TIM:

$$Q_{-j}^{[t_1, t_2]}(q) = \left\langle q(t_2), \Delta\theta_{-j}^{[t_2]} \right\rangle - \left\langle q(t_1), \Delta\theta_{-j}^{[t_1]} \right\rangle \tag{64}$$

where $\Delta\theta_{-j}^{[t]} = \theta_{-j}^{[t]} - \theta^{[t]}$.

Substituting $q(t) = \nabla_\theta \ell(z_{test}; \theta^{[t]})$ into Eq. (64):

$$Q_{-j}^{[t_1, t_2]}(q) = \left\langle \nabla_\theta \ell(z_{test}; \theta^{[t_2]}), \theta_{-j}^{[t_2]} - \theta^{[t_2]} \right\rangle - \left\langle \nabla_\theta \ell(z_{test}; \theta^{[t_1]}), \theta_{-j}^{[t_1]} - \theta^{[t_1]} \right\rangle. \tag{65}$$

Apply the first-order Taylor expansion of $\ell(z_{test}; \theta)$ around $\theta^{[t_2]}$ and $\theta^{[t_1]}$:

$$\ell(z_{test}; \theta_{-j}^{[t_2]}) \approx \ell(z_{test}; \theta^{[t_2]}) + \langle \nabla_\theta \ell(z_{test}; \theta^{[t_2]}), \theta_{-j}^{[t_2]} - \theta^{[t_2]} \rangle \tag{66}$$

$$\ell(z_{test}; \theta_{-j}^{[t_1]}) \approx \ell(z_{test}; \theta^{[t_1]}) + \langle \nabla_\theta \ell(z_{test}; \theta^{[t_1]}), \theta_{-j}^{[t_1]} - \theta^{[t_1]} \rangle \tag{67}$$

Rearranging Eq. (66) and Eq. (67):

$$\langle \nabla_\theta \ell(z_{test}; \theta^{[t_2]}), \theta_{-j}^{[t_2]} - \theta^{[t_2]} \rangle \approx \ell(z_{test}; \theta_{-j}^{[t_2]}) - \ell(z_{test}; \theta^{[t_2]}) \tag{68}$$

$$\langle \nabla_\theta \ell(z_{test}; \theta^{[t_1]}), \theta_{-j}^{[t_1]} - \theta^{[t_1]} \rangle \approx \ell(z_{test}; \theta_{-j}^{[t_1]}) - \ell(z_{test}; \theta^{[t_1]}) \tag{69}$$

Substituting these approximations back into Eq. (65):

$$Q_{-j}^{[t_1, t_2]}(q) \approx [\ell(z_{test}; \theta_{-j}^{[t_2]}) - \ell(z_{test}; \theta^{[t_2]})] - [\ell(z_{test}; \theta_{-j}^{[t_1]}) - \ell(z_{test}; \theta^{[t_1]})] \tag{70}$$

$$= [\ell(z_{test}; \theta_{-j}^{[t_2]}) - \ell(z_{test}; \theta_{-j}^{[t_1]})] - [\ell(z_{test}; \theta^{[t_2]}) - \ell(z_{test}; \theta^{[t_1]})] \tag{71}$$

This completes the proof of Theorem C.1. $\square$

This theorem provides a foundation for understanding how individual training data affects the model's loss on specific test points over time. The right-hand side of Eq. (63) represents the difference between the loss changes with and without data $z_j$, offering a direct measure of the data's influence on model performance.

**Extension to Test Sets:** We can extend this concept to consider an entire test set $D_{test} = \{z_1, \ldots, z_M\}$. Define the query function as:

$$q(t) = \frac{1}{M} \sum_{i=1}^{M} \nabla_\theta \ell(z_i; \theta^{[t]}), \quad z_i \in D_{test}. \tag{72}$$

With this choice, the TIM approximates the change in average test loss:

$$Q_{-j}^{[t_1, t_2]}(q) \approx \frac{1}{M} \sum_{i=1}^{M} \left[ \ell(z_i; \theta_{-j}^{[t_2]}) - \ell(z_i; \theta_{-j}^{[t_1]}) \right] - \frac{1}{M} \sum_{i=1}^{M} \left[ \ell(z_i; \theta^{[t_2]}) - \ell(z_i; \theta^{[t_1]}) \right]$$

$$= \left[ \mathcal{L}_{test}(\theta_{-j}^{[t_2]}) - \mathcal{L}_{test}(\theta_{-j}^{[t_1]}) \right] - \left[ \mathcal{L}_{test}(\theta^{[t_2]}) - \mathcal{L}_{test}(\theta^{[t_1]}) \right], \tag{73}$$

where $\mathcal{L}_{test}(\theta^{[t]}) = \frac{1}{M} \sum_{i=1}^{M} \ell(z_i; \theta^{[t]})$ is the average test loss.

## C.2 TIM FOR PREDICTION CHANGES

**Theorem C.2** (TIM for Prediction Changes). *Given a model function $f(x; \theta)$, a time window $[t_1, t_2]$, a training data $z_j$, and a test input $x_{test}$, the TIM with query function $q(t) = \nabla_\theta f(x_{test}; \theta^{[t]})$ can be approximated as:*

$$Q_{-j}^{[t_1, t_2]}(q) \approx \left[ f(x_{test}; \theta_{-j}^{[t_2]}) - f(x_{test}; \theta_{-j}^{[t_1]}) \right] - \left[ f(x_{test}; \theta^{[t_2]}) - f(x_{test}; \theta^{[t_1]}) \right], \quad (74)$$

*where $\theta_{-j}^{[t]}$ denotes the model parameters at time $t$ when trained without data $z_j$, and $\theta^{[t]}$ denotes the parameters when trained with all data.*

*Proof.* We begin with the definition of the query-based TIM:

$$Q_{-j}^{[t_1, t_2]}(q) = \left\langle q(t_2), \Delta\theta_{-j}^{[t_2]} \right\rangle - \left\langle q(t_1), \Delta\theta_{-j}^{[t_1]} \right\rangle \quad (75)$$

where $\Delta\theta_{-j}^{[t]} = \theta_{-j}^{[t]} - \theta^{[t]}$.

Substituting $q(t) = \nabla_\theta f(z_{test}; \theta^{[t]})$ into Eq. (75):

$$Q_{-j}^{[t_1, t_2]}(q) = \left\langle \nabla_\theta f(z_{test}; \theta^{[t_2]}), \theta_{-j}^{[t_2]} - \theta^{[t_2]} \right\rangle - \left\langle \nabla_\theta f(z_{test}; \theta^{[t_1]}), \theta_{-j}^{[t_1]} - \theta^{[t_1]} \right\rangle. \quad (76)$$

We apply the first-order Taylor approximation of the model function around $\theta^{[t_2]}$ and $\theta^{[t_1]}$:

$$f(x_{test}; \theta_{-j}^{[t_2]}) \approx f(x_{test}; \theta^{[t_2]}) + \langle \nabla_\theta f(x_{test}; \theta^{[t_2]}), \theta_{-j}^{[t_2]} - \theta^{[t_2]} \rangle \quad (77)$$

$$f(x_{test}; \theta_{-j}^{[t_1]}) \approx f(x_{test}; \theta^{[t_1]}) + \langle \nabla_\theta f(x_{test}; \theta^{[t_1]}), \theta_{-j}^{[t_1]} - \theta^{[t_1]} \rangle \quad (78)$$

Rearranging these equations:

$$\langle \nabla_\theta f(x_{test}; \theta^{[t_2]}), \theta_{-j}^{[t_2]} - \theta^{[t_2]} \rangle \approx f(x_{test}; \theta_{-j}^{[t_2]}) - f(x_{test}; \theta^{[t_2]}) \quad (79)$$

$$\langle \nabla_\theta f(x_{test}; \theta^{[t_1]}), \theta_{-j}^{[t_1]} - \theta^{[t_1]} \rangle \approx f(x_{test}; \theta_{-j}^{[t_1]}) - f(x_{test}; \theta^{[t_1]}) \quad (80)$$

Substituting these approximations back into Eq. (76):

$$Q_{-j}^{[t_1, t_2]}(q) \approx [f(x_{test}; \theta_{-j}^{[t_2]}) - f(x_{test}; \theta^{[t_2]})] - [f(x_{test}; \theta_{-j}^{[t_1]}) - f(x_{test}; \theta^{[t_1]})] \quad (81)$$

$$= [f(x_{test}; \theta_{-j}^{[t_2]}) - f(x_{test}; \theta_{-j}^{[t_1]})] - [f(x_{test}; \theta^{[t_2]}) - f(x_{test}; \theta^{[t_1]})] \quad (82)$$

This completes the proof of Theorem C.2. □

This theorem provides a formal justification for using TIM to analyze how excluding data $z_j$ influences the model's predictions on a test input $x_{test}$ over the interval $[t_1, t_2]$. Compared to Theorem C.1, which focuses on the loss value, Theorem C.2 focuses on specific model outputs. It enables the identification of influential training data for specific predictions, aids in understanding model functional response on particular inputs, and can help detect potential outliers or mislabeled data.

## C.3 TIM FOR FEATURE IMPORTANCE

**Theorem C.3** (TIM for Feature Importance). *Given a loss function $\ell(z; \theta)$, a training data $z = (x, y)$, and a test data $z_{test} = (x_{test}, y_{test})$, the TIM for feature importance with query function $q(t) = \nabla_x \nabla_\theta \ell(z_{test}; \theta^{[t]})$ can be approximated as:*

$$Q_{-j}^{[t_1, t_2]}(q) \approx [\nabla_x \ell(z_{test}; \theta_{-j}^{[t_2]}) - \nabla_x \ell(z_{test}; \theta_{-j}^{[t_1]})] - [\nabla_x \ell(z_{test}; \theta^{[t_2]}) - \nabla_x \ell(z_{test}; \theta^{[t_1]})], \quad (83)$$

*where $\theta_{-j}^{[t]}$ denotes the model parameters at time $t$ when trained without data $z_j$, and $\theta^{[t]}$ denotes the parameters when trained with all data.*

*Proof.* We start with the definition of the query-based TIM:

$$Q_{-j}^{[t_1,t_2]}(q) = \left\langle q(t_2), \Delta\theta_{-j}^{[t_2]} \right\rangle - \left\langle q(t_1), \Delta\theta_{-j}^{[t_1]} \right\rangle, \tag{84}$$

where $\Delta\theta_{-j}^{[t]} = \theta_{-j}^{[t]} - \theta^{[t]}$.

Substituting $q(t) = \nabla_x\nabla_\theta\ell(z_{\text{test}}; \theta^{[t]})$:

$$Q_{-j}^{[t_1,t_2]}(q) = \left\langle \nabla_\theta\nabla_x\ell(z_{\text{test}}; \theta^{[t_2]}), \theta_{-j}^{[t_2]} - \theta^{[t_2]} \right\rangle - \left\langle \nabla_\theta\nabla_x\ell(z_{\text{test}}; \theta^{[t_1]}), \theta_{-j}^{[t_1]} - \theta^{[t_1]} \right\rangle. \tag{85}$$

We apply the first-order Taylor approximation of $\nabla_x\ell(z_{\text{test}}; \theta)$ around $\theta^{[t_2]}$ and $\theta^{[t_1]}$:

$$\nabla_x\ell(z_{\text{test}}; \theta_{-j}^{[t_2]}) \approx \nabla_x\ell(z_{\text{test}}; \theta^{[t_2]}) + \nabla_\theta\nabla_x\ell(z_{\text{test}}; \theta^{[t_2]})\left(\theta_{-j}^{[t_2]} - \theta^{[t_2]}\right), \tag{86}$$

$$\nabla_x\ell(z_{\text{test}}; \theta_{-j}^{[t_1]}) \approx \nabla_x\ell(z_{\text{test}}; \theta^{[t_1]}) + \nabla_\theta\nabla_x\ell(z_{\text{test}}; \theta^{[t_1]})\left(\theta_{-j}^{[t_1]} - \theta^{[t_1]}\right). \tag{87}$$

Rearranging these equations:

$$\left\langle \nabla_\theta\nabla_x\ell(z_{\text{test}}; \theta^{[t_2]}), \theta_{-j}^{[t_2]} - \theta^{[t_2]} \right\rangle \approx \nabla_x\ell(z_{\text{test}}; \theta_{-j}^{[t_2]}) - \nabla_x\ell(z_{\text{test}}; \theta^{[t_2]}), \tag{88}$$

$$\left\langle \nabla_\theta\nabla_x\ell(z_{\text{test}}; \theta^{[t_1]}), \theta_{-j}^{[t_1]} - \theta^{[t_1]} \right\rangle \approx \nabla_x\ell(z_{\text{test}}; \theta_{-j}^{[t_1]}) - \nabla_x\ell(z_{\text{test}}; \theta^{[t_1]}). \tag{89}$$

Substituting these approximations back into Eq. (85):

$$Q_{-j}^{[t_1,t_2]}(q) \approx \left[\nabla_x\ell(z_{\text{test}}; \theta_{-j}^{[t_2]}) - \nabla_x\ell(z_{\text{test}}; \theta^{[t_2]})\right] - \left[\nabla_x\ell(z_{\text{test}}; \theta_{-j}^{[t_1]}) - \nabla_x\ell(z_{\text{test}}; \theta^{[t_1]})\right]$$

$$= \left[\nabla_x\ell(z_{\text{test}}; \theta_{-j}^{[t_2]}) - \nabla_x\ell(z_{\text{test}}; \theta_{-j}^{[t_1]})\right] - \left[\nabla_x\ell(z_{\text{test}}; \theta^{[t_2]}) - \nabla_x\ell(z_{\text{test}}; \theta^{[t_1]})\right]. \tag{90}$$

This completes the proof. $\qquad\square$

This theorem shows how TIM measures the impact of excluding a training data $z_j$ on the gradient of the loss with respect to the input features at the test point $z_{\text{test}}$ over the interval $[t_1, t_2]$. This provides insights into how the importance of different input features evolves during training and how individual training data influences this feature importance.

## C.4 TIM FOR PARAMETER IMPORTANCE

**Theorem C.4** (TIM for Parameter Importance). *Given a model with parameters $\theta \in \mathbb{R}^p$, a time window $[t_1, t_2]$, a training data $z_j$, and the $i$-th standard basis vector $e_i \in \mathbb{R}^p$, the TIM with query function $q(t) = e_i$ is exactly:*

$$Q_{-j}^{[t_1,t_2]}(q) = \left(\theta_{-j,i}^{[t_2]} - \theta_{-j,i}^{[t_1]}\right) - \left(\theta_i^{[t_2]} - \theta_i^{[t_1]}\right), \tag{91}$$

*where $\theta_{-j,i}^{[t]}$ denotes the $i$-th component of the model parameters at time $t$ when trained without data $z_j$, and $\theta_i^{[t]}$ denotes the $i$-th component of the parameters when trained with all data.*

*Proof.* We start with the definition of the query-based TIM:

$$Q_{-j}^{[t_1,t_2]}(q) = \left\langle q(t_2), \Delta\theta_{-j}^{[t_2]} \right\rangle - \left\langle q(t_1), \Delta\theta_{-j}^{[t_1]} \right\rangle, \tag{92}$$

where $\Delta\theta_{-j}^{[t]} = \theta_{-j}^{[t]} - \theta^{[t]}$.

Substituting $q(t) = e_i$, which is constant over time:

$$Q_{-j}^{[t_1,t_2]}(q) = \left\langle e_i, \theta_{-j}^{[t_2]} - \theta^{[t_2]} \right\rangle - \left\langle e_i, \theta_{-j}^{[t_1]} - \theta^{[t_1]} \right\rangle. \tag{93}$$

Since $e_i$ is the $i$-th standard basis vector, the inner product selects the $i$-th component:

$$Q_{-j}^{[t_1,t_2]}(q) = \left(\theta_{-j,i}^{[t_2]} - \theta_i^{[t_2]}\right) - \left(\theta_{-j,i}^{[t_1]} - \theta_i^{[t_1]}\right) = \left(\theta_{-j,i}^{[t_2]} - \theta_{-j,i}^{[t_1]}\right) - \left(\theta_i^{[t_2]} - \theta_i^{[t_1]}\right). \tag{94}$$

This matches the expression in Eq. (91), completing our proof. $\qquad\square$

This theorem allows us to isolate the influence of a training data $z_j$ on specific model parameters over the interval $[t_1, t_2]$. A large absolute value of $Q_{-j}^{[t_1,t_2]}(q)$ indicates that data $z_j$ has a significant influence on the $i$-th parameter during the specified time window. This is particularly useful for identifying which parameters are most affected by specific training data and understanding the localized effects of training data on the model.

By analyzing how $Q_{-j}^{[t_1,t_2]}(q)$ changes over different time windows, we can understand how the influence of training data on specific parameters evolves throughout the training process.

# D  IMPLEMENTATION OF TIM

## D.1  STANDARD IMPLEMENTATION OF TIM

Computing $Q_{-j}^{[t_1,t_2]}(q)$ for any query vector and training window without retraining is computationally attractive, but naive implementation faces significant challenges: 1) prohibitive storage overhead for tracking parameters and gradients across all training steps, 2) computational cost of Hessian matrix operations, and 3) complex influence propagation requiring intensive matrix calculations across multiple time steps.

We address these challenges with three technical innovations. For the storage challenge, we implement a selective window storage strategy that stores information only within user-specified windows $W$ during SGD training. To avoid costly Hessian computations, we employ Hessian-vector product Pearlmutter (1994)that eliminates the need for explicit Hessian matrices. For the third challenge, we develop a reverse-mode recursive propagation algorithm using auxiliary variables to track influence propagation without explicitly computing $\Delta\theta_{-j}^{[t_1,t_2]}$.

The implementation of TIM consists of two main algorithms: the data collection process during training (Algorithm 1) and the efficient influence computation (Algorithm 2).

**Model Training**  During standard SGD training, we strategically collect and store essential information $\{S_t, \eta_t, \theta^{[t+1]}\}$ required for subsequent influence analysis. As shown in Algorithm 1, this process is integrated seamlessly with standard training procedures while minimizing storage overhead.

The key feature is its selective storage strategy controlled by window $W$, which balances the period available for influence measurement and storage cost. The optimal $W$ depends on the task. Full-training storage is essential for optimizing curriculum learning schedules, while targeted windows covering convergence periods are sufficient for identifying corrupted data (see Table 6).

For scenarios with strict storage constraints, we design a checkpoint-based implementation (Appendix D.2) that greatly reduces storage to $O(Ep)$ while maintaining accuracy, $E$ is the steps per epoch, and $p$ is the parameter dimension.

---

**Algorithm 1** Standard Model Training

**Require:** Training dataset $D = \{z_n\}_{n=1}^N$, learning rate $\eta_t$, batch size $|S_t|$, training steps $T$, selectable storage window $W$
**Ensure:** Stored information $A$
1: Initialize model parameters $\theta^{[0]}$
2: Initialize an empty sequence $A$
3: **for** $t = 0$ to $T - 1$ **do**
4:    $S_t = \text{DataBatch}(D, |S_t|)$
5:    $\theta^{[t+1]} = \theta^{[t]} - \frac{\eta_t}{|S_t|}\sum_{i\in S_t}g(z_i;\theta^{[t]})$
6:    **if** $t \in W$ **then**
7:       $A[t] = \{S_t, \eta_t, \theta^{[t+1]}\}$
8:    **end if**
9: **end for**
10: **return** $A$

---

**Algorithm 2** TIM Data Influence Computation

**Require:** Stored training information $A$, query function $q$, time window $[t_1, t_2]$, specified data $z_j$
**Ensure:** The influence $Q$ of data $z_j$
1: Initialize $Q \leftarrow 0$, $u_1^{[t_2-1]} \leftarrow 0$
2: Initialize $u_2^{[t_2-1]} \leftarrow q(t_2)$
3: **for** $t = t_2 - 1$ **downto** $0$ **do**
4:    **if** $j \in S_t$ **then**
5:       $Q \leftarrow Q + \left\langle (u_2^{[t]} - u_1^{[t]}), \frac{\eta_t}{|S_t|}g(z_j;\theta^{[t]}) \right\rangle$
6:    **end if**
7:    $u_1^{[t-1]} \leftarrow u_1^{[t]} - \eta_t H^{[t]}u_1^{[t]}$
8:    $u_2^{[t-1]} \leftarrow u_2^{[t]} - \eta_t H^{[t]}u_2^{[t]}$
9:    **if** $t = t_1$ **then**
10:       $u_1^{[t-1]} \leftarrow q(t_1)$
11:    **end if**
12: **end for**
13: **return** $Q$

---

**Influence Computation**  Algorithm 2 implements the computation of query-based influence $Q_{-j}^{[t_1,t_2]}(q)$ in Eq. (11) using the stored training information. The algorithm employs a reverse-time

recursive propagation approach that avoids explicitly computing the parameter influence $\Delta\theta_{-j}^{[t_1,t_2]}$, which would be prohibitively expensive for large models.

The algorithm uses $u_1^{[t]}$ and $u_2^{[t]}$, which represent how earlier parameter changes propagate to the query directions $q(t_1)$ and $q(t_2)$, respectively. They are recursively updated in reverse time by multiplying with $P_t$, which models how parameter changes propagate through the optimization trajectory. When $z_j$ appears in mini-batch $S_t$, its gradient is projected onto $u_2^{[t]} - u_1^{[t]}$, capturing its relative influence at that step. Appendix E provide a formal proof that Algorithm 2 correctly computes $Q_{-j}^{[t_1,t_2]}(q)$ as defined in Eq. (11).

TIM avoids computing Hessian matrices directly, which would require $O(Tp^2)$ operations. $p$ is the parameter dimension, and $T$ is the number of training steps. Instead, it uses efficient Hessian-vector products $H^{[t]}u = \nabla_\theta \langle u, g(z; \theta^{[t]}) \rangle$ Pearlmutter (1994), reducing cost to $O(|S_t|p)$ per step.

## D.2  CHECKPOINT-BASED IMPLEMENTATION OF TIM

To balance storage overhead and computational efficiency, we propose a checkpoint-based implementation of TIM. This implementation significantly reduces storage requirements while maintaining the ability to compute accurate influence values for any time window.

Instead of storing parameters at every training step, we store checkpoints at regular intervals (e.g., epoch boundaries) along with essential training metadata (batch indices and learning rates). When computing influence for a time window $[t_1, t_2]$, we efficiently recover necessary parameters by loading the nearest checkpoint before $t_1$, reconstructing the parameter trajectory up to $t_2$, and storing intermediate parameters required for influence computation. The checkpoint interval provides a configurable trade-off between storage overhead and computational cost. More frequent checkpoints reduce recomputation but increase storage, while fewer checkpoints save storage at the cost of more recomputation.

---

**Algorithm 3** Training with Checkpoints

**Require:** Training dataset $D = \{z_n\}_{n=1}^N$, learning rate $\eta_t$, batch size $|S_t|$, training steps $T$, checkpoint interval $C$
**Ensure:** Stored checkpoints and metadata $M$
1: Initialize model parameters $\theta^{[0]}$
2: Initialize metadata storage $M \leftarrow \{\}$ {Store checkpoints, batch indices, learning rates}
3: **for** $t = 0$ **to** $T - 1$ **do**
4:   $S_t \leftarrow \text{DataBatch}(D, |S_t|)$
5:   $\theta^{[t+1]} \leftarrow \theta^{[t]} - \frac{\eta_t}{|S_t|} \sum_{i \in S_t} g(z_i; \theta^{[t]})$
6:   $M.indices[t] \leftarrow S_t$ {Store batch indices}

7:   $M.lr[t] \leftarrow \eta_t$ {Store learning rate}
8:   **if** $t \bmod C = 0$ **or** $t = T - 1$ **then**
9:     $M.checkpoints[t] \leftarrow \theta^{[t+1]}$ {Store checkpoint}
10:   **end if**
11: **end for**
12: **return** $M$

---

**Algorithm 4** TIM Data Influence with Checkpoints

**Require:** Metadata $M$, query function $q$, time window $[t_1, t_2]$, data $z_j$
**Ensure:** Estimated influence $Q$
1: $Q \leftarrow 0$, $u_1^{[t_2-1]} \leftarrow 0$, $u_2^{[t_2-1]} \leftarrow q(t_2)$
2: $c_1 \leftarrow \max\{t : t \leq t_1 \text{ and } t \in M.checkpoints\}$ {Find nearest checkpoint before $t_1$}
3: $\theta^{[c_1]} \leftarrow M.checkpoints[c_1]$
4: {Compute and store all necessary parameters from checkpoint to $t_2$}
5: Initialize parameter storage $P \leftarrow \{\}$
6: **for** $t = c_1$ **to** $t_2 - 1$ **do**
7:   $S_t \leftarrow M.indices[t]$
8:   $\eta_t \leftarrow M.lr[t]$
9:   **if** $t \in M.checkpoints$ **then**
10:     $\theta^{[t]} \leftarrow M.checkpoints[t]$
11:   **end if**
12:   $\theta^{[t+1]} \leftarrow \theta^{[t]} - \frac{\eta_t}{|S_t|} \sum_{i \in S_t} g(z_i; \theta^{[t]})$
13:   $P[t] \leftarrow \theta^{[t]}$ {Store parameter for influence computation}
14: **end for**
15: **for** $t = t_2 - 1$ **downto** $t_1$ **do**
16:   **if** $j \in M.indices[t]$ **then**
17:     $Q \leftarrow Q + \left\langle (u_2^{[t]} - u_1^{[t]}), \frac{M.lr[t]}{|M.indices[t]|} g(z_j; P[t]) \right\rangle$
18:   **end if**
19:   $H^{[t]} \leftarrow \frac{1}{|M.indices[t]|} \sum_{i \in M.indices[t]} \nabla_\theta g(z_i; P[t])$
20:   $u_1^{[t-1]} \leftarrow u_1^{[t]} - M.lr[t] H^{[t]} u_1^{[t]}$
21:   $u_2^{[t-1]} \leftarrow u_2^{[t]} - M.lr[t] H^{[t]} u_2^{[t]}$
22:   **if** $t = t_1$ **then**
23:     $u_1^{[t-1]} \leftarrow q(t_1)$
24:   **end if**
25: **end for**
26: **return** $Q$

---

## E  PROOF OF ALGORITHM 2

We begin by recalling the definition:

$$Q_{-j}^{[t_1,t_2]}(q) = \langle q(t_2), \Delta\theta_{-j}^{[t_2]}\rangle - \langle q(t_1), \Delta\theta_{-j}^{[t_1]}\rangle \tag{95}$$

where $\Delta\theta_{-j}^{[0,t]} \approx \sum_{s=0}^{t-1}\left(\prod_{k=s+1}^{t-1}P_k\right)\tilde{\mathbf{1}}_j^{[s]}$, and $P_t = I - \eta_t H^{[t]}$, $\tilde{\mathbf{1}}_j^{[t]} = \mathbf{1}_{j\in S_t}\frac{\eta_t}{|S_t|}g(z_j;\theta^{[t]})$.

Note that $P_t$ is self-adjoint matrix, adhering to $\langle x, P_t y\rangle = \langle P_t x, y\rangle$ for all vectors $x, y$.

According to the update rules for $u_1$ and $u_2$ in the algorithm:

$$u_i^{[t-1]} = u_i^{[t]} - \eta_t H^{[t]}u_i^{[t]} = (I - \eta_t H^{[t]})u_i^{[t]} = P_t u_i^{[t]}, \quad i \in \{1,2\} \tag{96}$$

By recursive application of this update rule, we obtain for $s < t$:

$$u_i^{[s]} = \left(\prod_{k=s+1}^{t-1}P_k\right)u_i^{[t]}, \quad i \in \{1,2\} \tag{97}$$

According to the accumulation of $Q$ in the algorithm, at each time step $t$, if $j \in S_t$, we have:

$$\Delta Q_t = \left\langle (u_2^{[t]} - u_1^{[t]}), \frac{\eta_t}{|S_t|}g(z_j;\theta^{[t]})\right\rangle \tag{98}$$

The algorithm initializes $u_2^{[t_2-1]} = q(t_2)$ and sets $u_1^{[t_1-1]} = q(t_1)$ at time $t_1$. Importantly, $u_1$ is not updated beyond $t_1$. Using the result from Eq. (97), we can express $u_2^{[t]}$ and $u_1^{[t]}$ as:

$$u_2^{[t]} = \prod_{k=t+1}^{t_2-1}P_k q(t_2), \quad \text{for } 0 \le t < t_2 \tag{99}$$

$$u_1^{[t]} = \begin{cases}\prod_{k=t+1}^{t_1-1}P_k q(t_1) & \text{for } 0 \le t < t_1 \\ 0 & \text{for } t_1 \le t < t_2\end{cases} \tag{100}$$

Note that $u_1^{[t]} = 0$ for $t_1 \le t < t_2$ because $u_1$ is not updated beyond $t_1$, effectively removing its contribution to $\Delta Q_t$ in this range.

Substituting these expressions into Eq. (98):

$$\Delta Q_t = \begin{cases}\left\langle \prod_{k=t+1}^{t_2-1}P_k q(t_2) - \left(\prod_{k=t+1}^{t_1-1}P_k q(t_1)\right), \tilde{\mathbf{1}}_j^{[t]}\right\rangle & \text{for } 0 \le t < t_1 \\ \left\langle \prod_{k=t+1}^{t_2-1}P_k q(t_2), \tilde{\mathbf{1}}_j^{[t]}\right\rangle & \text{for } t_1 \le t < t_2\end{cases} \tag{101}$$

The total $Q$ is the sum of all $\Delta Q_t$: $Q = \sum_{t=0}^{t_2-1}\Delta Q_t$.

Expanding this sum and recalling that $P_t$ is self-adjoint, we get:

$$Q = \left\langle q(t_2), \sum_{t=0}^{t_2-1}\left(\prod_{k=t+1}^{t_2-1}P_k\right)\tilde{\mathbf{1}}_j^{[t]}\right\rangle - \left\langle q(t_1), \sum_{t=0}^{t_1-1}\left(\prod_{k=t+1}^{t_1-1}P_k\right)\tilde{\mathbf{1}}_j^{[t]}\right\rangle \tag{102}$$

Note that $u_2^{[t]}$ contributes to the first term over the entire interval $[0, t_2)$, while $u_1^{[t]}$ only contributes to the second term over $[0, t_1)$. This distinction arises from the algorithm's design, where $u_1$ is not updated beyond $t_1$.

Combined Eq. (102) are precisely the definitions of $\Delta\theta_{-j}^{[t_2]}$ and $\Delta\theta_{-j}^{[t_1]}$, we have:

$$Q = \langle q(t_2), \Delta\theta_{-j}^{[t_2]}\rangle - \langle q(t_1), \Delta\theta_{-j}^{[t_1]}\rangle = Q_{-j}^{[t_1,t_2]}(q) \tag{103}$$

Thus, we have rigorously demonstrated that the algorithm's output $Q$ is equivalent to the defined $Q_{-j}^{[t_1,t_2]}(q)$ in Eq. (95) under the stated assumption on $\eta_t$.

# F    EXPERIMENTAL SUPPLEMENT

## F.1    EXPERIMENTAL SETUP

Experiments were conducted on eight NVIDIA RTX A5000 GPUs (24GB each), dual Intel Xeon Gold 6342 CPUs (2.80 GHz, 96 cores), and 503GB RAM. Implementation uses Ubuntu 22.04.3 LTS, PyTorch v2.4.1, CUDA 12.4, and Python 3.11.9. All results are reported as mean $\pm$ standard deviation over 16 runs with different random seeds.

**Datasets**    We employed four diverse datasets spanning various domains and complexities to evaluate the robustness and generalizability of TIM.

- **Adult** Dua & Graff (2019): A tabular dataset with 48,842 instances and 14 features.
- **20 Newsgroups** Lang (1995): A text classification dataset. Text data is converted to TF-IDF vectors, and stop words are removed for cleaner feature representation.
- **IMDB Movie Reviews** Maas et al. (2011): A sentiment analysis dataset containing 50,000 movie reviews with binary sentiment labels (positive/negative). Reviews are tokenized using WordPiece tokenization and truncated to a maximum sequence length of 512 tokens.
- **MNIST** LeCun et al. (2010): An image dataset with 70,000 grayscale images across 10 classes. We use a binary task distinguishing digits '1' and '7'. Each image is $28 \times 28$ pixels and normalized.
- **EMNIST** Cohen et al. (2017): An image dataset containing 131,600 images across 47 classes. Each image is $28 \times 28$ pixels and is normalized for consistency.

**Model Architectures**    We evaluated TIM using different model architectures of varying complexity.

- **BERT** Devlin et al. (2019): For sentiment analysis on IMDB, we use BERT-base-uncased as the pre-trained model with 110 million parameters. The model consists of 12 transformer layers with 768 hidden dimensions and 12 attention heads.
- **Vision Transformer (ViT)**: A compact vision transformer model with approximately 1.8 million parameters. Vision transformer adopts a multi-layer transformer architecture with self-attention and MLP blocks, introducing substantial depth and non-linearity. Unlike CNNs, its global receptive field and parameter-sharing across layers make optimization highly non-convex.
- **Convolutional Neural Network (CNN)**: This architecture is used for image datasets like MNIST and EMNIST. It consists of two convolutional layers, with 32 and 64 filters, respectively, each followed by ReLU activation and max-pooling. The final output from the convolutional layers is flattened and passed through a linear layer to output a binary classification value.
- **Logistic Regression (LR)**: Implemented as a single-layer neural network without hidden layers. The input dimension is flattened to accommodate various input shapes.
- **Deep Neural Network (DNN)**: The architecture comprises two hidden layers, each with eight units followed by a ReLU activation function. The second layer outputs a single value for binary classification. The input is flattened, similar to logistic regression.

For non-image data like Adult and 20 Newsgroups, the input is a vector, while image data like MNIST and EMNIST is reshaped into a single dimension for LR and DNN models. The CNN processes image data in its original 2D format. All these models output a single value and use binary cross-entropy loss with logits for classification, with input/output dimensions adapted to each dataset.

**Compared Methods**    We compare TIM against the following influence measurement methods.

- **Leave-One-Out (LOO)** serves as ground truth, measuring influence by retraining without data $z_j$. $\Delta\ell_{LOO}(z_j) = \frac{1}{M}\sum_{i=1}^{M}(\ell(z_i, \theta_{-j}) - \ell(z_i, \theta))$, where $z_i \in D_{\text{test}}$, $M$ is the size of the test set $D_{\text{test}} = \{z_i\}_{i=1}^{M}$.

- **Influence Functions (IF)** Koh & Liang (2017) estimates the influence of removing a training data $z_j$ on the model's overall loss for a test set $D_{\text{test}}$: $I(z_j, D_{\text{test}}) = -\frac{1}{M} \sum_{i=1}^{M} \nabla_\theta \ell(z_i, \theta)^T H^{-1} \nabla_\theta \ell(z_j, \theta)$, where $H$ is the Hessian of the model's loss at $\theta$.

- **TracIn** Pruthi et al. (2020): $\text{TracIn}(z_j, z_i) = \sum_{k=1}^{K} \eta_k \nabla \ell(\theta^{[k]}, z_j) \cdot \nabla \ell(\theta^{[k]}, z_i)$, where $\theta^{[k]}$ is checkpoints of model parameters.

- **Lava** Just et al. (2023): measures influence through optimal transport cost gradients between training and validation datasets. The influence of training point $(x_i, y_i)$ is quantified as: $\phi_{\text{LAVA}}(x_i, y_i) := h_i^* - \frac{1}{n-1} \sum_{j \in [n] \setminus \{i\}} h_j^*$, where $(h_1^*, \ldots, h_n^*)$ is part of the optimal dual solution for the optimal transport problem between training and validation distributions.

- **DVEmb** Wang et al. (2025b) Estimates influence via an inner product $\text{DVEmb}(z_j, z_i) \approx v_j^T \nabla_\theta \ell(z_i, \theta)$, where $v_j \in \mathbb{R}^d$ is a low-dimensional vector. The embedding $v_j$ is updated recursively at each step of the training trajectory to capture temporal dynamics.

- **TIM** measures influence by setting $q(t) = \frac{1}{M} \sum_{i=1}^{M} \nabla_\theta \ell(z_i; \theta^{[t]})$, measuring the impact on test set $D_{\text{test}}$ loss across time window $[t_1, t_2]$: $Q_{-j}^{[t_1, t_2]}(q) \approx \frac{1}{M} \sum_{i=1}^{M} \left[ \ell(z_i; \theta_{-j}^{[t_2]}) - \ell(z_i; \theta_{-j}^{[t_1]}) \right] - \frac{1}{M} \sum_{i=1}^{M} \left[ \ell(z_i; \theta^{[t_2]}) - \ell(z_i; \theta^{[t_1]}) \right]$.

**Evaluation Metrics**  To comprehensively evaluate the performance of TIM, we employed a suite of statistical metrics, each capturing different aspects of the relationship between the compared methods:

- **Pearson Correlation Coefficient ($r$)** Pearson (1895): The Pearson correlation coefficient measures the linear relationship between two variables. For two sets of data, X and Y, it is calculated as:

$$r = \frac{\sum_{i=1}^{n} (X_i - \bar{X})(Y_i - \bar{Y})}{\sqrt{\sum_{i=1}^{n} (X_i - \bar{X})^2 \sum_{i=1}^{n} (Y_i - \bar{Y})^2}}$$

where $\bar{X}$ and $\bar{Y}$ are the means of X and Y respectively, and $n$ is the number of data points. This metric is valuable for identifying direct proportional or inversely proportional relationships within the data. $r$ ranges from -1 to 1, where 1 indicates a perfect positive linear relationship, -1 indicates a perfect negative linear relationship, and 0 indicates no linear relationship.

- **Spearman's Rank Correlation Coefficient ($\rho$)** Spearman (1987): Spearman's rank correlation assesses monotonic relationships by comparing the rank orders of data points:

$$\rho = 1 - \frac{6 \sum_{i=1}^{n} d_i^2}{n(n^2 - 1)}$$

where $d_i$ is the difference between the ranks of corresponding values $X_i$ and $Y_i$, and $n$ is the number of data points. $\rho$ ranges from -1 to 1, with values close to 1 or -1 indicating strong monotonic relationships (positive or negative, respectively) and values close to 0 indicating weak monotonic relationships.

- **Kendall's Tau ($\tau$)** Kendall (1938): Kendall's Tau evaluates ordinal relationships by measuring the number of concordant and discordant pairs:

$$\tau = \frac{2(n_c - n_d)}{n(n - 1)}$$

where $n_c$ is the number of concordant pairs, $n_d$ is the number of discordant pairs, and $n$ is the total number of pairs. $\tau$ ranges from -1 to 1, with 1 indicating perfect agreement between two rankings, -1 indicating perfect disagreement, and 0 indicating no relationship.

- **Jaccard Similarity ($J$)** Jaccard (1912): The Jaccard similarity coefficient compares the overlap between the top 30% of influential points as determined by different methods:

$$J(A, B) = \frac{|A \cap B|}{|A \cup B|}$$

where $A$ and $B$ are the sets of top 30% influential points identified by different methods. $J$ ranges from 0 to 1, with 1 indicating perfect overlap between the sets and 0 indicating no overlap.

By capturing linear relationships (Pearson), monotonic relationships (Spearman), ordinal relationships (Kendall's Tau), and set-based similarities (Jaccard), we ensure a multifaceted evaluation of influence analysis methods.

To ensure transparency and reproducibility, all code, including detailed hyperparameter settings and training procedures, is available on our GitHub repository https://anonymous.4open.science/r/TIM-DE8E/. This repository contains scripts and configuration files that define the exact setup for each model used in our experiments, encompassing learning rates, batch sizes, regularization strategies, and any other model-specific training details.

### F.2 METHOD OF DATA INFLUENCE DYNAMICS

To investigate how the influence of individual training data evolves, we conduct a systematic analysis using LOO as ground truth. The method for analyzing data influence dynamics consists of the following steps:

1. **Influence Tracking**: We randomly select 256 training data points and track their influence. For each selected data point $z_j$, we compute its LOO influence on test loss at every epoch by comparing the standard model trained on the complete dataset and a modified model trained with identical settings but excluding $z_j$. The LOO influence is quantified as the difference in test loss between these two models. By repeating this measurement across all training epochs, we can directly observe how each data point's influence on model performance evolves over time, revealing dynamic influence patterns that static methods cannot capture.

2. **Standardization**: We standardize the influence values separately within each epoch using scikit-learn's StandardScaler, which transforms values to have zero mean and unit variance using the formula $z = \frac{x-\mu}{\sigma}$, where $x$ is the original influence value, $\mu$ is the mean influence across all data points at that epoch, and $\sigma$ is the standard deviation. This epoch-wise standardization preserves relative influence relationships while removing the global declining scale effect.

3. **Time-Varying Pattern Categorization**: For each data point, a linear regression is performed on its standardized influence values over time. The slope of this regression line indicates the overall trend direction (increasing or decreasing influence). The p-value of the regression determines whether this trend is statistically significant. Training data are categorized based on their statistical properties, including a) Trend significance (determined by the p-value) b) Trend direction (positive or negative slope) c) Standard deviation of influence values (a measure of fluctuation).

4. **Pattern Analysis**: We calculate the proportion of data in each category and compute the centroid of each category by averaging the standardized influence values of all data within that category. These centroids represent the typical trend of each pattern and are plotted in Figure 4 to visually demonstrate the characteristics of each influence pattern. We also report the distribution of patterns across datasets and model architectures in Table 4, showing that influence dynamics vary significantly depending on both model and data modality.

### F.3 ADDITIONAL ANALYSIS FOR SECTION 5.1

We evaluate the accuracy of TIM in measuring data influence on test loss by comparing it against IF, using LOO as ground truth. Since IF operates only on the final model, we use TIM with a full training window to match its global influence scope and ensure a fair comparison. We report four agreement metrics with LOO: Pearson and Spearman correlations for linear and monotonic relationships, respectively, Kendall's tau for ordinal relationships, and Jaccard similarity for the top 30% influencers. Detailed metric descriptions are in Appendix F.1. Qualitative scatterplots are deferred to Appendix F.3.

Table 7 shows several key findings. First, TIM consistently surpasses IF in accuracy across all datasets and model architectures, achieving correlations of up to 0.99 (Pearson and Spearman) for LR models. Second, TIM's advantage is most significant in complex settings like non-convex DNN and MNIST, where it maintains high correlations while IF's performance drops significantly. Third, TIM shows superior robustness and reliability, with lower standard deviations (typically $\pm 0.01$) across runs compared to IF (up to $\pm 0.33$).

Table 7: Comparison of TIM and IF accuracy against LOO across models and datasets. Higher is better. Means and standard deviations are over 16 random seeds.

| Model | Dataset | Pearson | | Spearman | | Kendall's Tau | | Jaccard | |
|---|---|---|---|---|---|---|---|---|---|
| | | TIM | IF | TIM | IF | TIM | IF | TIM | IF |
| LR | Adult | **0.99±0.01** | 0.91±0.04 | **0.99±0.01** | 0.93±0.02 | **0.95±0.01** | 0.79±0.04 | **0.91±0.04** | 0.71±0.06 |
| | 20News | **0.99±0.01** | 0.90±0.13 | **0.99±0.01** | 0.94±0.08 | **0.97±0.01** | 0.84±0.13 | **0.95±0.03** | 0.78±0.16 |
| | MNIST | **0.93±0.10** | 0.76±0.14 | **0.98±0.01** | 0.61±0.22 | **0.95±0.02** | 0.49±0.21 | **0.91±0.05** | 0.48±0.14 |
| DNN | Adult | **0.95±0.02** | 0.88±0.04 | **0.95±0.03** | 0.86±0.04 | **0.83±0.06** | 0.69±0.05 | **0.75±0.08** | 0.56±0.07 |
| | 20News | **0.85±0.07** | 0.77±0.05 | **0.85±0.08** | 0.80±0.06 | **0.71±0.08** | 0.62±0.07 | **0.67±0.08** | 0.55±0.07 |
| | MNIST | **0.96±0.03** | 0.75±0.14 | **0.94±0.06** | 0.70±0.17 | **0.83±0.08** | 0.52±0.14 | **0.78±0.15** | 0.52±0.19 |

Furthermore, we conducted a pattern-specific accuracy analysis comparing TIM against IF. For each dataset–model pair, we compute the per-example influence on test loss using TIM (full-window) and IF, and compare it against LOO retraining as ground truth. Each point is a training example; the $x$-axis is the LOO loss difference, and the $y$-axis is the estimated loss difference from TIM (blue) or IF (red). The dashed line denotes $y = x$ (perfect agreement).

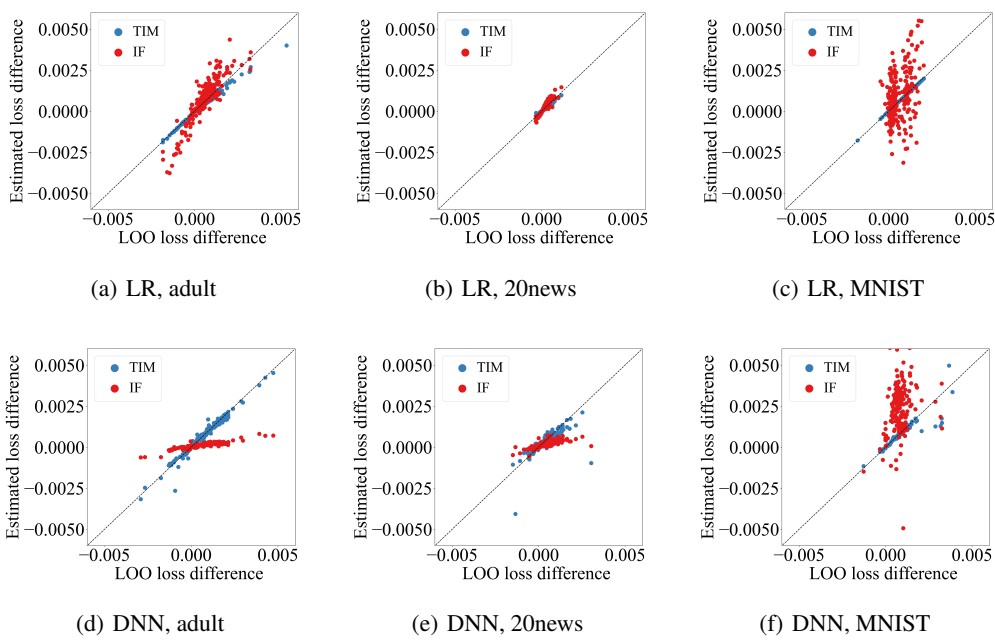

(a) LR, adult      (b) LR, 20news      (c) LR, MNIST

(d) DNN, adult      (e) DNN, 20news      (f) DNN, MNIST

Figure 5: TIM and IF influence measurements compared to LOO ground truth. The x-axis shows LOO values; the y-axis shows the measured influence from TIM (blue) and IF (red). Points closer to the diagonal ($y = x$) indicate higher accuracy.

TIM consistently aligns more closely with the $y = x$ line than IF, indicating better alignment with the ground truth. This advantage is particularly pronounced in complex, non-convex deep learning settings.

### F.4 INFLUENCE DYNAMICS AND SIMILARITY ACROSS TRAINING STAGES

After validating TIM's accuracy in estimating data influence, we used it to analyze the similarity of different training stages. The training process was adaptively divided into early, middle, and late stages using change points identified in the overall training loss trajectory. Specifically, we modeled the training loss using an exponential decay curve to capture the overall trend and reduce noise. This approach helps to smooth out fluctuations and emphasize underlying trends in the training

loss. Then, we compute residuals as the differences between the actual loss values and the values predicted by the exponential model. These residuals highlight where the actual training deviates from the predicted trend. Third, we identified peaks in the absolute residuals as change points. A minimum distance criterion was applied to ensure these change points were evenly distributed across the training timeline. Finally, based on the identified change points, the training process was divided into three stages: early, middle, and late. We set time windows based on stages and used TIM to compute data influence within these windows. We then used Kendall's tau correlation to quantify the similarity of influence rankings between stages, with higher values indicating greater stability. Table 8 presents these correlations.

Table 8: Kendall's Tau correlations across training stages across datasets and models

| Model | Dataset | Early-Middle | Early-Late | Middle-Late | Early-Full | Middle-Full | Late-Full |
|---|---|---|---|---|---|---|---|
| LR | Adult | $0.64 \pm 0.14$ | $0.62 \pm 0.08$ | $0.79 \pm 0.14$ | $0.81 \pm 0.05$ | $\mathbf{0.82 \pm 0.12}$ | $0.79 \pm 0.05$ |
| | 20News | $0.79 \pm 0.12$ | $0.78 \pm 0.10$ | $0.79 \pm 0.09$ | $\mathbf{0.91 \pm 0.02}$ | $0.88 \pm 0.10$ | $0.86 \pm 0.12$ |
| | MNIST | $0.43 \pm 0.14$ | $0.15 \pm 0.12$ | $0.35 \pm 0.14$ | $0.71 \pm 0.08$ | $\mathbf{0.72 \pm 0.09}$ | $0.30 \pm 0.14$ |
| | EMNIST | $0.73 \pm 0.04$ | $0.40 \pm 0.16$ | $0.51 \pm 0.18$ | $0.83 \pm 0.03$ | $\mathbf{0.89 \pm 0.02}$ | $0.49 \pm 0.17$ |
| DNN | Adult | $0.61 \pm 0.11$ | $0.41 \pm 0.15$ | $0.70 \pm 0.06$ | $0.7 \pm 0.09$ | $\mathbf{0.87 \pm 0.04}$ | $0.69 \pm 0.08$ |
| | 20news | $0.66 \pm 0.06$ | $0.57 \pm 0.07$ | $0.76 \pm 0.05$ | $0.81 \pm 0.03$ | $\mathbf{0.82 \pm 0.04}$ | $0.76 \pm 0.04$ |
| | MNIST | $0.56 \pm 0.06$ | $0.18 \pm 0.21$ | $0.20 \pm 0.25$ | $0.74 \pm 0.03$ | $\mathbf{0.81 \pm 0.04}$ | $0.20 \pm 0.25$ |
| | EMNIST | $0.60 \pm 0.12$ | $0.40 \pm 0.20$ | $0.59 \pm 0.21$ | $0.69 \pm 0.11$ | $\mathbf{0.84 \pm 0.07}$ | $0.63 \pm 0.17$ |

Table 8 shows several key insights. First, data influence evolves significantly throughout training, as evidenced by the consistently low correlations between early and late stages (Early-Late column). This challenges the static influence measurement methods and highlights the necessity for time-aware methods like TIM. Second, mid-training influence strongly correlates with full-training influence across all datasets and models. This suggests that influential data can be identified before convergence. Mid-training analysis can approximate full-training data influence, potentially reducing computational costs. These insights have significant implications for data selection and curriculum learning strategies. Third, for a given dataset, the patterns of influence ranking changes at different stages are similar across different model architectures when accounting for standard deviations. This consistency suggests that the influence of data is largely determined by the inherent dataset rather than being heavily model-dependent.

### F.5 SCALABILITY TO VIT

To evaluate TIM's scalability, we compare TIM and TracIn using a Vision Transformer (ViT). This setting significantly exceeds prior influence analysis work in model complexity. We compare TIM against TracIn Pruthi et al. (2020), a representative method for large-scale, non-convex models. Traditional approaches such as IF and LOO are excluded due to their prohibitive computational cost at this scale.

We evaluate corruption detection capability by randomly flipping 2%, 4%, 6%, and 8% of training labels (160, 320, 480, and 640 corrupted data points, respectively). For each scenario, we train the ViT model on the corrupted dataset, compute influence scores using last-epoch TIM and TracIn, and rank data points by their negative influence.

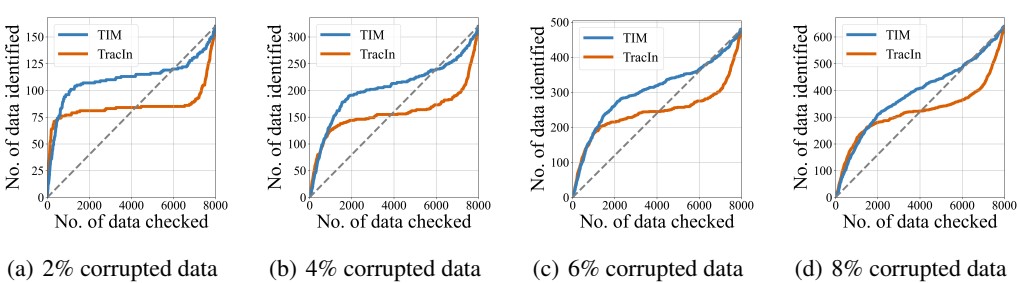

(a) 2% corrupted data     (b) 4% corrupted data     (c) 6% corrupted data     (d) 8% corrupted data

Figure 6: Comparison of TIM and TracIn for corrupted data detection on EMNIST using ViT.

Figure 6 shows that TIM consistently outperforms TracIn, achieving higher detection accuracy by identifying more corrupted data when examining the same number of training data. These results confirm that TIM scales effectively to modern deep architectures and complex datasets, providing reliable influence analysis beyond existing methods.

### F.6 Accelerating Convergence

Data influence analysis can accelerate model convergence through strategic data pruning. We evaluated this on an MNIST classification task (distinguishing between digits '1' and '7') using a DNN with 30% flipped labels. We compared three strategies: 1) training with corrupted data; 2) full-training TIM prune, which removes the bottom 30% influential data points based on global influence measured over the entire training trajectory; and 3) per-epoch TIM prune, which dynamically removes the bottom 30% influential data at each epoch.

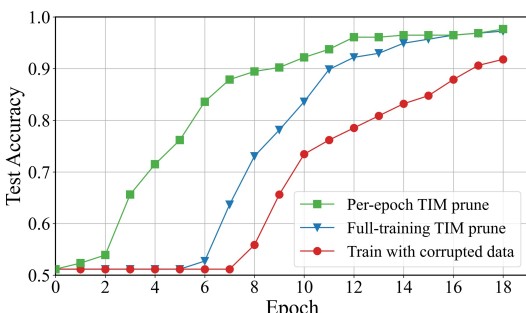

Figure 7: Comparison of model convergence rates with different pruning strategies on MNIST-DNNs.

As shown in Figure 7, per-epoch TIM pruning achieves 85% accuracy within six epochs, far ahead of other methods. This reveals key benefits of time-varying influence measurement. First, TIM enables significantly faster convergence by pruning data at each epoch. Second, the performance gap between per-epoch TIM and full-training TIM pruning validates our finding that data influence patterns evolve throughout training, making window-specific analysis superior to global influence measurement. Third, TIM can be used as an adaptive curriculum learning approach, automatically identifying optimal training data for each epoch without requiring manual curriculum design.

## LLM Usage Disclosure

We used Large Language Models (LLMs) in limited ways during the preparation of this work. Specifically, LLMs were employed to polish the language for clarity and conciseness, rephrase sections to better match the academic style expected in machine learning venues, and assist in exploring potentially relevant related work by suggesting references and keywords for further manual inspection. All conceptual contributions, methodological innovations, theoretical analyses, and experimental designs were conceived and validated solely by the human authors. Similarly, all implementations, data analyses, and reported results were conducted and verified by the authors. Suggested related works from LLMs were cross-checked manually to ensure correctness.

