# OpenReview forum: "How Data Influence Changes in Training? Time-Varying Influence Measurement"
_ICLR.cc/2026/Conference — ICLR 2026 Conference Withdrawn Submission_

### Official Review · Reviewer_QrW6 · 2025-10-25

**Soundness:** 3
**Presentation:** 3
**Contribution:** 3
**Rating:** 6
**Confidence:** 5

**Summary:**

This paper introduces Time-varying Influence Measurement, a novel methads for quantifying how the influence of individual training examples evolves over thw course of model training. Unlike traditional influence analysis methods (e.g., Influence Functions, Shapley values), which provide only a static, global estimate of data influence on the final model, TIM operates over arbitrary local time windows during training. It estimates the effect of removing a data point within a window on model parameters and then projects this deviation onto task-relevant functional responses (e.g., test loss) using query vectors. The authors provide theoretical error bounds under non-convex and non-converged settings, and demonstrate through extensive experiments that TIM:

1. Accurately approximates Leave-One-Out (LOO) retraining with high correlation across multiple metrics;

2. Reveals distinct temporal influence patterns (e.g., Early Influencers, Late Bloomers);

3. Enables more efficient and effective corrupted data detection and convergence acceleration—often with 95% less computation than full-trajectory analysis.

**Strengths:**

**1. Novelty and Timeliness:**
TIM is the first framework to explicitly model time-varying data influence during training. This addresses a critical gap in the literature, where most influence methods assume a fixed, converged model.

**2. Strong Theoretical Foundation:**
The paper provides a rigorous error analysis (Appendix B) that holds without requiring convexity or convergence—key limitations of classical Influence Functions. This makes the method applicable to modern deep learning settings.

**3. Practical Efficiency and Scalability:**
TIM leverages Hessian-vector products and reverse-mode propagation to avoid explicit Hessian computation or full retraining. The checkpoint-based implementation further reduces storage overhead, enabling application to large models like BERT and ViT.

**Weaknesses:**

1. The paper treats query vectors (e.g., test loss gradient) as given, but doesn’t explore how the choice of query affects influence interpretation or robustness. A deeper discussion on query sensitivity or adaptive query selection would strengthen the framework.

2. Corrupted data experiments use random label flipping, which may not reflect real-world data quality issues (e.g., systematic bias, domain shift). Testing on more realistic noise or dataset errors would better demonstrate practical utility.

3. Besides, in my personal opinion, the layout and the aesthetic appeal of the pictures in this article really need to be improved!

**Questions:**

See Weakness

---

### Official Review · Reviewer_VtYD · 2025-10-31

**Soundness:** 2
**Presentation:** 2
**Contribution:** 2
**Rating:** 2
**Confidence:** 4

**Summary:**

This paper introduces Time-varying Influence Measurement (TIM), a framework intended to measure how the influence of individual data points changes throughout the model training process, rather than as a single static value. The method aims to approximate Leave-One-Out (LOO) retraining over arbitrary time windows by tracking parameter deviations. The paper compares the proposed method with several notable methods for data evaluation and conducts empirical studies with a subset of these methods. The paper claims that the results show the proposed approach yields scores that best correlate with LOO-measured influence.

While the problem of understanding dynamic data influence is interesting and important, the paper's core innovation and contribution are unclear. Presentation is flawed--both conception discussions and empirical comparisons did not include the most relevant methods/baselines. The implementations of baseline methods are questionable. The paper may have exaggerated its contributions and overclaimed on several issues.

**Strengths:**

1. The paper tackles the relevant and challenging problem of measuring data influence dynamically, moving beyond the limitations of static, end-of-training analysis.

2. Comparisons and discussions are comprehensive: covering theoretical aspects, empirical comparisons in several setups, and ablation studies.

**Weaknesses:**

1. The method propagates influence using an "Influence Propagator" ($P_t = I - \eta_t H^{[t]}$), which relies on Hessian computing. Do we need to compute this for every step within the local window?

2. The paper claims (Table 1) a computational cost of $O(w|S_t|p)$, which is linear in the number of parameters $p$, but the "Recursive Estimation" involves computing Hessian matrices, which have $O(p^2)$ costs. It is not explained how these are connected.

3. The paper asserts that TIM "addresses" non-convexity, but the update mechanism is based on Hessian propagation, which is a local, second-order approximation. It is an overclaim that TIM has overcome global non-convexity challenges

4. Crucial baselines that also track influence throughout training are missing from both conceptual discussions, which affects how this work contextualizes.

5. The experimental comparisons are questionable. TIM should be compared against other methods tracking influence throughout training (or aggregated influences for TracIn or IF estimated from multiple checkpoints).

6. Experiment setup and implementation details are missing. LAVA results appear inconsistent with those reported in other works. It is unclear whether LAVA was implemented on raw features, BERT embeddings, or embeddings fine-tuned on validation data.


7. For the BERT-IMDB experiment (Section 5.2). This experiment uses a binary sentiment classification task with 50% randomly flipped labels. If 50% of the labels in a binary task are flipped, the labels become entirely random and uncorrelated with the data. It is confusing. Is this a valid task?

**Questions:**

1. Can the authors provide a clear derivation for the $O(w|S_t|p)$ complexity? Specifically, how is the product of Influence Propagators (Eq. 10) computed in a way that is only linear in $p$?

2. Please explain the setup of the BERT-IMDB experiment. How should we interpret the binary classification task with 50% label noise?

3. Please provide the precise implementation details for the methods/baselines in the experiments. For the LAVA baseline, what are the features/embeddings used in the experiments?

4. Why is LAVA not considered an approximation of LOO? Can the author offer some more elaborations on this?

---

### Official Review · Reviewer_5oDQ · 2025-11-02

**Soundness:** 2
**Presentation:** 3
**Contribution:** 2
**Rating:** 4
**Confidence:** 3

**Summary:**

This paper introduces Time-varying Influence Measurement (TIM), a framework designed to measure how the influence of individual training data points evolves within arbitrary windows of the training process. The method's core is a recursive estimation of parameter deviations caused by data removal within a specified training window. These parameter changes are then projected onto behavior-specific query vectors (e.g., the test loss gradient) to quantify the impact on multiple dimensions of model behavior, such as test loss, predictions, and feature importance. The paper establishes theoretical error bounds for this estimation under non-convex and non-converged conditions and empirically identifies several temporal influence patterns, such as Early Influencers and Late Bloomers.

**Strengths:**

1. The paper's framework for measuring data influence dynamically is an interesting and important direction in this field. It effectively shifts the paradigm from a static, post-hoc analysis to a dynamic, process-oriented one.

2. The method's strong empirical correlation with leave-one-out (LOO) retraining is impressive, demonstrating TIM's superior accuracy over other baselines.

3. The identification of temporal patterns, such as Early Influencers and Late Bloomers, offers a valuable analytical lens for understanding when and how models learn from different data subsets, improving the interpretability of the training process.

**Weaknesses:**

1.  My main concern is the insufficient validation of the method's practical utility. Specifically, the early detection experiments in Table 3 are post-hoc analyses, not on-the-fly evaluations. I wonder how accurately other baselines would detect bad data on the fly, for example, at 3 out of 10 epochs. Does a performance gap still exist in this setting? And even then, can TIM achieve similar accuracy by only looking at the [2, 3] window? Furthermore, there are many ways to scale up IF in the current literature. Why do the authors use different baselines across different models for bad data detection?

2.  I wonder if these "Late Bloomer" and "Early Influencer" patterns still occur in large-scale models. The authors claim their method is more scalable than other approaches. If so, why do they not include experiments in a large-scale setting where this analysis would be more useful? If the specific use case for this analysis is curriculum learning, I wonder if these dynamics would even be observable in LLM-scale training. For example, can a single sample really show such dynamics, or would this only be meaningful at a subset level?

3.  Another concern is the claim of multi-dimensional influence analysis. This is positioned as one of the contributions, but it is not substantiated with empirical evidence. The experiments rely on test loss as the evaluation metric. While Appendix C provides the theory, the paper would be stronger if it included at least one experiment in the main paper to demonstrate the practical utility of this claimed flexibility.

**Questions:**

Please see the details in the Weakeness

---

### Official Review · Reviewer_WrXo · 2025-11-05

**Soundness:** 2
**Presentation:** 2
**Contribution:** 1
**Rating:** 2
**Confidence:** 5

**Summary:**

The paper proposes TIM, a framework to measure how the influence of a training point changes over time by operating on arbitrary training windows. TIM first estimates the parameter deviation caused by excluding a point within the window via a recursive “influence propagator”, and then projects this deviation onto a query vector to obtain changes in a chosen functional (e.g., test loss). Experimental results show that TIM provides accurate estimation to LOO, identifies distinct temporal influence patterns, and delivers practical benefits including improved corrupted-data detection and faster convergence.

**Strengths:**

The need to measure the influence of training data across different training phases is well‑motivated.

**Weaknesses:**

**Literature Review and Related Work**

SGD-influence (Hara et al., 2019) has spawned several follow-up works, yet the paper only cites and compares against DVEmb. Important works are missing from the comparison, including:
- Bae, Juhan, et al. "Training data attribution via approximate unrolling." NeurIPS 2024
- Zhang, Shichang, et al. "Accountability Attribution: Tracing Model Behavior to Training Processes." arXiv preprint arXiv:2506.00175 (2025).

I recommend consulting the related work sections in DVEmb and the above papers for a more thorough literature review.

**Novelty**

The derivation in Section 4.1 should explicitly acknowledge that it is largely adapted from the standard SGD-influence derivation from Hara et al. (2019). If I understand correctly, the primary distinction between SGD-influence and the proposed technique is that this work examines data point appearances within a specific time window. This is a relatively incremental contribution, and the connection to the line of SGD-influence research is completely unacknowledged in the paper. In the Related Work section, the authors' characterization of SGD-influence/DVEmb as work that "lacks theoretical analysis and shows poor experimental results" is not appropriate. For example, both papers (as well as the uncited papers in this line) have a similar theoretical analysis to the one in this paper. Please carefully revise the related work section and explicitly acknowledge the role of the proposed technique in the line of SGD-influence literature. Currently, the contribution of this work seems to be significantly overclaimed.

**Efficiency**
- The approach requires users to specify the time window for attribution beforehand, which may be impractical in many scenarios.
- The algorithm in Appendix D.1 appears to be a simplified version of DVEmb, whereas DVEmb proposes several techniques to reduce computational cost.
- The runtime of Algorithm 2 actually exceeds model retraining when $t_2 = T$ and $t_1 = 0$ (global analysis) due to the additional Hessian computation overhead.

Table 1 contains several unclear elements:
- What does $C_{\text{train}}$ represent?
- What is the difference between $d$ and $\tilde p$?
- The $O(p^3)$ complexity for IF seems outdated given the extensive literature on reducing this complexity.
- The $O(p^2)$ storage cost for IF is unclear, even in the naive setting.

I suggest separating compute costs into offline and online components, as demonstrated in Appendix C.9 of [1] and Table 3 of [2]:

[1] Wang, Jiachen T., et al. "Capturing the temporal dependence of training data influence." ICLR 2025
[2] Deng, Junwei, et al. "A Survey of Data Attribution: Methods, Applications, and Evaluation in the Era of Generative AI." (2025).

**Experiment**

At the algorithmic level, the primary differences between DVEmb and the proposed technique appear to be that DVEmb uses the Fisher Information Matrix (FIM) to estimate the Hessian and employs random projection to reduce computational and storage costs. Given these relatively minor differences, it is surprising that DVEmb shows low correlation with ground-truth LOO in Table 2. What projection dimension was used for DVEmb in these experiments?

For local window analysis, the statement "we construct their local estimates by differencing the loss between [0, $t_2$] and [0, $t_1$]" requires clarification. If I understand correctly, DVEmb and Zhang et al., (2025) can be readily adapted to local window analysis, so the claimed advantage is unclear.

**Presentation**

Please use `\citep` for parenthetical in-text citations rather than `\cite`.

**Questions:**

See weakness.

---

### Note · Authors · 2025-11-26

I have read and agree with the venue's withdrawal policy on behalf of myself and my co-authors.